# AVATREE: An open-source computational modelling framework modelling Anatomically Valid Airway TREE conformations

**Stavros Nousias** [ORCID]***, Evangelia I. Zacharaki, Konstantinos Moustakas**

Department of Electrical and Computer Engineering, University of Patras, Patras, Greece

* snousias@upatras.gr

**Data Availability Statement:** The data underlying the results presented in the study are available from https://vessel12.grand-challenge.org/. The repository is now publicly available at

## Abstract

This paper presents AVATREE, a computational modelling framework that generates Anatomically Valid Airway tree conformations and provides capabilities for simulation of broncho-constriction apparent in obstructive pulmonary conditions. Such conformations are obtained from the personalized 3D geometry generated from computed tomography (CT) data through image segmentation. The patient-specific representation of the bronchial tree structure is extended beyond the visible airway generation depth using a knowledge-based technique built from morphometric studies. Additional functionalities of AVATREE include visualization of spatial probability maps for the airway generations projected on the CT imaging data, and visualization of the airway tree based on local structure properties. Furthermore, the proposed toolbox supports the simulation of broncho-constriction apparent in pulmonary diseases, such as chronic obstructive pulmonary disease (COPD) and asthma. AVATREE is provided as an open-source toolbox in C++ and is supported by a graphical user interface integrating the modelling functionalities. It can be exploited in studies of gas flow, gas mixing, ventilation patterns and particle deposition in the pulmonary system, with the aim to improve clinical decision making.

## 1 Introduction

In the past years, a multitude of studies paves the way for the generation of patient-specific computational models of lung structure and function. Early studies focused on airway morphometry generating the first human bronchial trees models [1]. These studies employed casts to decipher the relationship between bronchi lengths, branching angles and airway diameters [2]. On this basis, researchers built and validated a simulation model of airway morphogenesis from generation 1 to generation 23 [3, 4]. Deterministic parameterized bronchial tree generation algorithms used as single input the location of the first one or two generations and the lung volume, extracted directly from computed tomography, thus constituting the core of patient-specific modelling [5–7]. Personalized boundary conditions based on diagnostic imaging were combined with generative approaches and lumped models of resistive trees [8, 9] constituting the state of the art in pulmonary system modelling. Later studies incorporated

https://gitlab.com/LungModelling/avatree. Furthermore, the outcomes of the presented pipeline are available at https://www.kaggle.com/vvrlabeceupatras/pone-avatree-results.

**Funding:** This work has been co-financed by the European Regional Development Fund of the European Union and Greek national funds through the Operational Program Competitiveness, Entrepreneurship and Innovation, under the call RESEARCH – CREATE - INNOVATE Take-A-Breath, under grant agreement No. T1EDK-03832). The funders had no role in study design, data collection and analysis, decision to publish, or preparation of the manuscript.

**Competing interests:** The authors have declared that no competing interests exist.

patient-specific boundary conditions into computational fluid dynamics to examine flow regimes, wall stresses and aerosol deposition. In the same direction, modelling the airflow in cases of constrictive conditions, such as asthma and chronic obstructive pulmonary disease (COPD) became feasible with the aforementioned approaches. Wall constriction and remodelling combined with patient-specific boundary conditions allowed the quantification of breathing conditions for asthmatic patients. Motivated by these advancements we introduce an end-to-end modelling approach that produces Anatomically Valid Airway tree conformations (AVATREE). Such conformations are adapted to personalized geometry and boundary conditions derived from diagnostic imaging and well-established airway extraction methods. Specifically, this study aims to provide an open-source simulation framework to (i) exploit imaging data so as to provide patient-specific representations (ii) perform structural analysis (iii) extend the segmented airway tree to predict the airway branching across the whole lung volume (iv) visualize probabilistic confidence maps of generation data (v) simulate bronchoconstriction to (vi) access patient-specific airway functionality (vii) perform fluid dynamics simulation in patient-specific boundary conditions to access pulmonary function.

## 1.1 Background & related work

While early studies focused mainly on quantitative modeling approaches to gain insight into the lung function without an explicit link to the lung's structure, with the advancements in computing power and the current medical imaging capabilities, the interest in the simulation of lung function based on personalized geometric models that incorporate the essential structural features of the lungs, has significantly increased [10]. By now, many studies propose the development and adoption of mathematical and geometrical models to study the structure of the airways and pulmonary physiology. Some address the problem of airway tree segmentation from CT images, while others analyze the branching patterns and bifurcations through airway morphometry or mathematical modelling. In this section, we briefly present representative approaches and introduce definitions for the different computational steps required during airway and pulmonary structure modelling and simulation. Airway segmentation, bronchial morphometry and tree branching, mathematical models of bifurcating distributing systems are required to derive patient-specific structural and functional modelling approaches.

Early studies on airway morphometry [11–13] used casts of human lungs to study branching patterns and the relation between airway lengths and diameters. The most commonly used conducting airway model has been Weibel's symmetric model "A" [14]. The airway position has also been described by Horsfield order [1] and Strahler order [15]. Later on, with the advancement of medical imaging techniques, the extraction of airway structure and lung volume from imaging started to play an important role in the analysis of pulmonary diseases. A literature review on the analysis of lung CTs, including segmentation of the various pulmonary structures, can be found in [16], while a comparative study of automated and semi-automated segmentation methods of the airway tree from CT images was presented in [17]. Overall, segmentation approaches can be classified into methods based on morphology [18], morphological aggregation [19], voxel classification [20], adaptive region growing with constraints [21–28], tube similarity [29, 30] and gradient vector flow [31]. Several implementations of the aforementioned approaches are available in the literature. The tube segmentation framework [30] utilizing gradient vector flow [31] and the FAST heterogeneous medical image computing and visualization framework [32] utilizing the seeded region growing approach. AVATREE employs airway segmentation as a first step to obtain the personalized structure in the first generations, while the more advanced generations are simulated based on a tree extension algorithm.

Furthermore, mathematical models of the airway structure were formulated to derive branching and structural rules. Deterministic mathematical models of bifurcating distributing systems were examined [3] setting the basis for modelling bronchial tree branching as a function of available lung space [4]. Deriving airway diameter as a relation of branching features facilitates full determination of the geometry given skeletal representations. Several studies mention scaling properties [33–36] for the airway diameters so that the average diameter $D$ of a given airway at generation $G$ is the product of the diameter of the trachea $D_0$. Furthermore, Kamiya et al. [37] validated the relation between airway diameter and branching angles and, Kitaoka et al. [2] proposed a branching model allowing the prediction of the relationship between branching angle and flow rate and between airway length and diameter. Experimental studies verified the validity of the aforementioned methods. For the surface reconstruction of airway surface Tawhai et al. [5, 6, 38] employed fitting cubic Hermite surfaces as described in [39]. The study of Hegedus et al. [40] generated surface models of idealized bifurcation through mathematical modelling rigorously extending the previous definitions [41]. The aforementioned studies are relevant to our approach. To avoid the definition of special rules in the reconstruction of the surface of bifurcations we define the same boundary conditions, as the Poisson-reconstructed surface of a sampled point cloud.

Towards patient-specific structural and functional modelling, Tawhai et al. and Lin et al. [6, 42] studied the imposition of patient-specific boundary conditions to generate 1-dimensional and three-dimensional computational models taking into consideration the effects of turbulence. Towards the same direction, a review article [10] provides insight into multiscale finite element models of lung structure and function aiming towards a computational framework for bridging the spatial scales from molecular to the whole organ. Bordas et al. [7] developed an image analysis and modelling pipeline applied to healthy and asthmatic patient scans to produce complete personalized airway models to the acinar level incorporating CT acquisition, lung and lobar segmentation, airway segmentation and centerline extraction, algorithmic generation of distal airways and zero-dimensional models. Their implementation and results were included into Chaste framework [43], an open-source framework to facilitate computational modelling in heart, lung and soft tissue simulations. Towards the same direction, Montesantos et al. [44] presented a detailed algorithm for the generation of an individualized 3D deterministic model of the conducting part of the human tracheobronchial tree. With respect to the aforementioned studies, our work focuses on generating surface meshes of extended patient-based bronchial trees, suitable for computational fluid dynamics (CFD) simulations, along with a toolbox to simulate constriction of the airways.

Several authors employed CFD to investigate flow regimes in the human lung. In our previous work [45, 46] we performed narrowing deformations in CT extracted lung geometries to simulate constrictive conditions. Other studies in the same category include simulations for CT-based patient specific geometries [47–52], particle deposition [53–56], constrictive pulmonary diseases [45, 46, 57–59], micro-airway flow regimes [57], turbulence modelling [60], four-dimensional (space and time) dynamic simulations [61], ventilation heterogeneity [57], airflow in the acinar region [62]. Validation studies conducted by Montesantos et al. [63] include morphometric studies on healthy and asthmatic patients providing among others, measurements of branching angles, length and diameter of airways as a function of generation. Such measurements are employed by our study for macroscopic validation of the generated trees.

## 1.2 Motivation and contributions

The objective in this field of research is to enable the prediction of gas flow [51, 55], gas mixing [64], heat transfer [65], particle deposition [46, 54, 66, 67], and ventilation distribution [68] in

the pulmonary system. Lung ventilation patterns prediction [69, 70] can provide grounds for performance and fatigue estimation in high-frequency ventilation cases [71], disease severity quantification, such as in asthma and COPD, and give insight into drug delivery or even in transfer of harmful particulates. Motivated by the recent advances in this field and building upon previous work [46], we developed an end-to-end approach facilitating pulmonary structural modelling that is based on the definition of the personalised boundary conditions required for fluid dynamics simulations. Specifically, in this work we

- present an open-source simulation framework that utilizes imaging data to provide patient-based representations of the structural models of the bronchial tree,

- build and extend 1-dimensional graph representations of the bronchial tree,

- generate 3D surface models of extended bronchial tree models appropriate for CFD simulations

- generate probabilistic visualization of airway generations projected on the personalized CT imaging data, 2

- perform validation studies and provide comparison with relevant state-of-the-art approaches

- provide an open-source toolbox in C++ and a graphical user interface integrating modelling functionalities.

The rest of the paper is organized as follows. Section 2 analyzes the individual components of AVATREE, Section 3 commends on the results of our approach while Section 4 concludes this paper.

## 2 Materials and methods

The processing pipeline uses as input CT images and is presented in Fig 1. Airway segmentation is applied to the imaging data to extract a 3D surface mesh and a 1-dimensional representation of the airways. We employ the extended 1D graph to derive visualization of probabilistic airway generation labels in the space of the subjects' anatomy as defined by the CT images and to generate a 3D surface defining personalized boundary conditions, that can be employed as input for computational fluid dynamics simulations.

### 2.1 Segmentation and airways centerline extraction

The input of the presented approach is unlabeled CT scans required to extract bronchial tree and airways structural features. For the definition of the lung volume, CT-based lung segmentation and annotation is required. For lung segmentation we employ the FAST heterogeneous medical image computing and visualization framework [32] is employed. The result of lung segmentation process is a binary mask visualized in Fig 1. As a next step, we perform further processing of the segmentation result to distinguish left and right lungs. The process is described below:

1. A second region growing takes place starting from a single random point inside any of the segmented region only if all its immediate neighbours bare the same label.

2. To advance the region growing front, all points neighbouring a candidate voxel must not include background voxel. This region is given a new label.

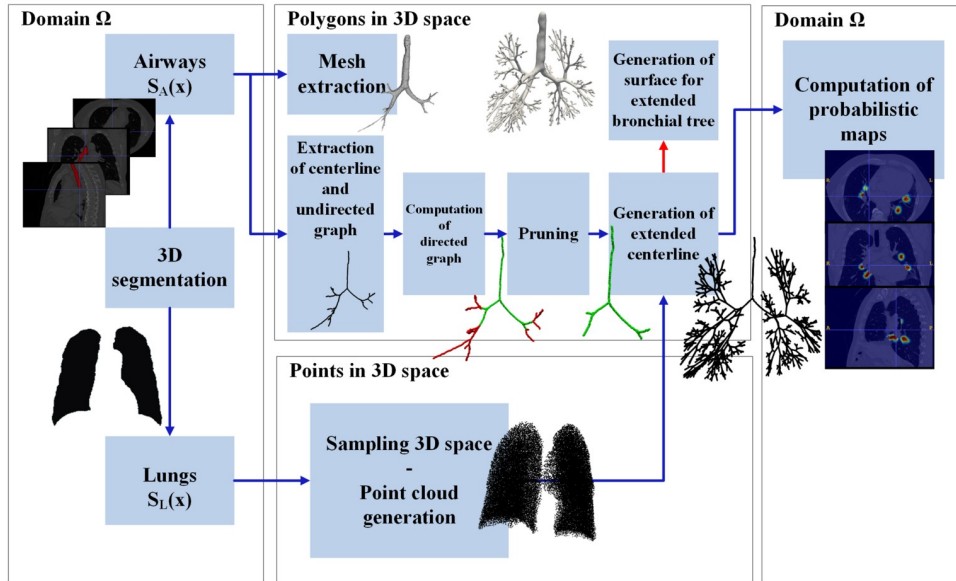

**Fig 1. Processing pipeline of AVATREE.**

3. Steps 1 and 2 are repeated for the other lung volume. The result is an image with three labels(background and two lung volumes).

4. To distinguish left or right we employ the directed graph extracted from the main airways and follow the generic rule according to which the topological distance the topological distance between the bifurcations of the first and the second generation is longer in the left lung.

The next step involves the segmentation of the first generations of the airways that are identifiable in the patient's CT image, but any available airway tree segmentation method can be also applied. For this purpose we investigated two algorithms. The first algorithm is the gradient vector flow [29, 31] which achieved high accuracy with low false-positive rate (only 1.44%) in a comparative study [17] in the context of the EXACT09 airway segmentation challenge. The second is a standard and stable approach based on seeded region growing [72]. The former is included in the tube segmentation framework [30] and the latter in FAST heterogeneous medical image computing and visualization framework [32].

Let's denote with $I(\mathbf{x})$, $I : \Omega \rightarrow \mathcal{R}$, the gray level 3D medical image, where $\mathbf{x} = (x, y, z)$, $\mathbf{x} \in \Omega$ is a voxel in the spatial domain $\Omega \subset \mathcal{R}^3$ of the volumetric imaging data. The output of the segmentation algorithm for the airways is a binary image $S_A$ of equal size with $I$. Likewise, the output of the segmentation algorithm for the lung volumes is a binary image $S_L$ of equal size with $I$. The result is presented in Figs 1 and 2, and utilized to generate prediction of full bronchial tree structures based on personalized lung volumes. To derive the centerline from $S_A$ a multitude of methods is provided in the literature including skeletonization or thinning. Fig 2 presents the up-to-four generations centerline of the airways.

This 1D representation of the bronchial tree is modelled by an undirected graph $\mathcal{G} = \{\mathcal{V}, \mathcal{E}\}$ where $\mathcal{V}$ is the set of vertices and $\mathcal{E}$ is the set of edges. Each vertex, indexed by $i$, can be represented as a point $\mathbf{v}_i = (x_i, y_i, z_i)$. We denote the function $N(\mathbf{v}_i)$ yielding the set of vertices indices neighbouring vertex $i$. The undirected graph is extracted by FAST framework [32] and converted into a directed graph with the following process. Initially, the graph starting point is

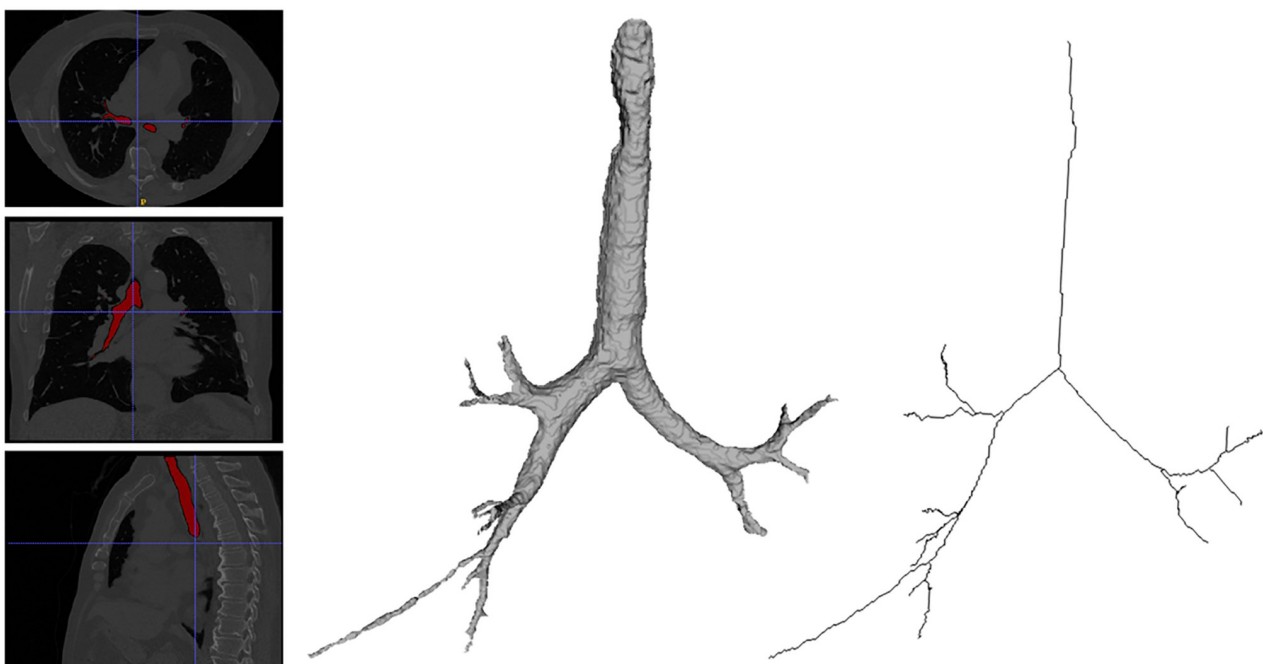

**Fig 2. Extraction of airway surface and centerline.**

defined as the one closest to the air inlet, i.e. the oral cavity or the trachea. Given index $y$ the starting point for $\mathcal{G}$ we generate the directed tree $\mathcal{G}_D$. We define as distal point the vertex of the graph with no children and distal branch the edge containing a distal point.

**Algorithm 1** Detection of inlet in undirected graph

```
Input: Graph 𝒢
Output: Index of graph inlet y
1 procedure DERIVATION OF GRAPH INLET
2    Initialize set 𝒫 = [||]
3    for each vertex vᵢ do
4       if |N(vᵢ)| > 2 then
5          for each n ∈ N(vᵢ) do
6             Initialize empty set 𝒦 = {i, n}
7             while N(vₙ)<3 do
8                for each m ∈ N(vₙ) do
9                   if m∉𝒫 then 𝒦 ← 𝒦 ∪ m
10            𝒫 ← 𝒫 ∪ 𝒦
11   𝒦_max = max_{1≤i≤|𝒫|}Length(𝒦ᵢ)
12   y ← 𝒦_{max_{|𝒦_max|}}
```

## 2.2 Generation of extended bronchial tree

Since higher generations cannot be identified from the personal imaging data, we extend the bronchial tree based on population-wise empirical observations. Initially the directed graph generated by the procedure explained in subsection 2.1 is pruned. Specifically, the extracted 1-dimensional representation is processed to include all the bifurcations located at the end of a given generation so as to facilitate the volume filling algorithm. Fig 1 shows the result of pruning where all generations after the $n^{th}$ have been pruned. The corrected tree is subsequently used for the bronchial tree extension. The generation process utilizes the bronchial tree

extension algorithm initially proposed by Tawhai et al. [4] and later improved by Bordas et al. [7] while introducing a few safeguards to allow maximal space utilization. The bronchial tree extension algorithm can be described by the following steps.

For each lung subvolume $S_{L_L}$ and $S_{L_R}$:

1. Generate a point cloud sampling the subvolume with a uniform random process. Fig 3 depicts the uniform sampling of each lung subvolume with a total number of $n = 30000$ points [4, 6].

2. Assign a seed point to the closest distal branch as presented in Fig 3.

3. Calculate the center of mass of the sampled points as presented in Fig 3.

$$\mathbf{c} = \frac{\sum_{\mathbf{p}_i \in \mathcal{P}} \mathbf{p}_i}{|\mathcal{P}|} \qquad (1)$$

4. Employ principal component analysis (PCA) on the set of sampled points to define the splitting plane. The motivation for employing PCA is to address a space utilization aspect. The direction of the eigenvector with the greatest norm indicates the dimension of the data with the greatest variance denoting the direction where more space is available for the branches to grow. Picking a plane so that the resulting bounding box demonstrates the lowest possible variation, inhibits the appearance of very long branches. Given data points $\mathbf{D} = [\mathbf{p}_1 \ \mathbf{p}_1 \ \mathbf{p}_1 \cdots \mathbf{p}_n]$, $\mathbf{A} = \mathbf{D}\mathbf{D}^T$ is the auto-correlation matrix. Direct singular value decomposition yields $\mathbf{A} = \mathbf{U}\mathbf{\Sigma}\mathbf{U}^T$ where $\mathbf{U} = [\mathbf{u}_1 \ \mathbf{u}_2 \ \mathbf{u}_3]$. Then the largest eigenvector is defined as $\mathbf{u}_m = \max_{1 \le i \le 3} \mathbf{u}_i$. Given the vector $\mathbf{d}$ expressing the direction of the distal airway, the splitting plane is described by center of mass $\mathbf{c}$ and vector $\mathbf{d} \times [\mathbf{d} \times \mathbf{u}_m]$. The selected plane maximizes the available space for each new subdivision. Fig 4 presents a splitting plane splitting the set of points into two subdivisions.

5. Calculate the centroid of each new subdivision.

6. For each centroid define line segment starting from seed point extending 40% of the distance towards centroid of the subdivision.

7. If a newly created branch is smaller that $2mm$, it is considered as terminal.

8. The process is repeated until no seed points remain.

9. Any branch found outside the lung volume is removed along with children branches.

It is important to denote that the presented pipeline enables the generation of a tunable user-defined number of generations. If $n$ is the number of desired generations, we set stopping criteria, in the extension of the bronchial tree until $2^{(n+1)}$ bifurcations have been reached. The resulting 1D representation (Fig 5) predicts the location of the bifurcating distributive [3] structure given the patient-specific available space. The outcome of the volume filling algorithm will be used later to create maps that express the probability of a voxel to belong to a certain generation. This information when projected on CT slices can be a very informative and powerful clinical decision support tool.

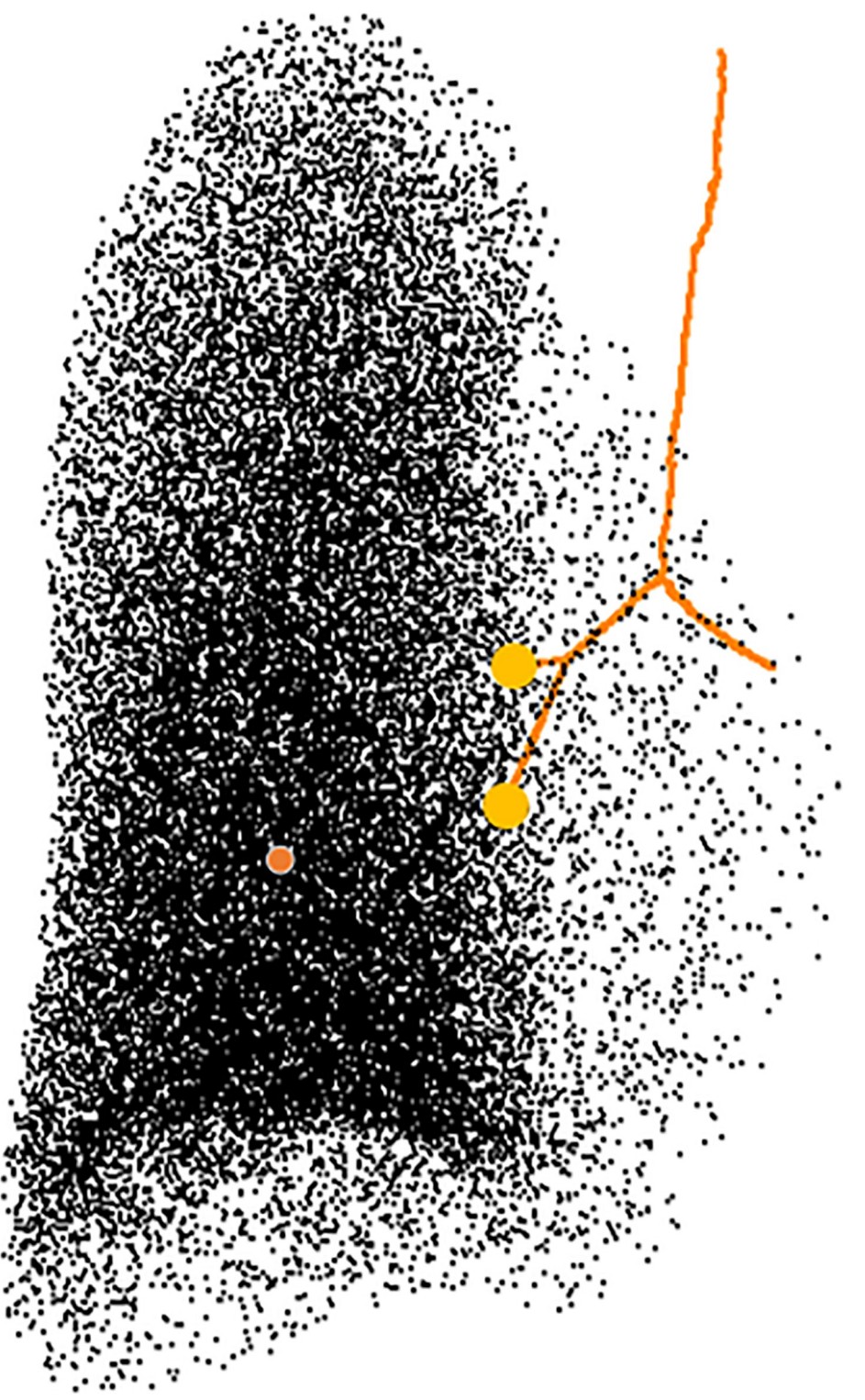

**Fig 3. Definition of center of mass (orange dot) and distal branches (yellow dots).**

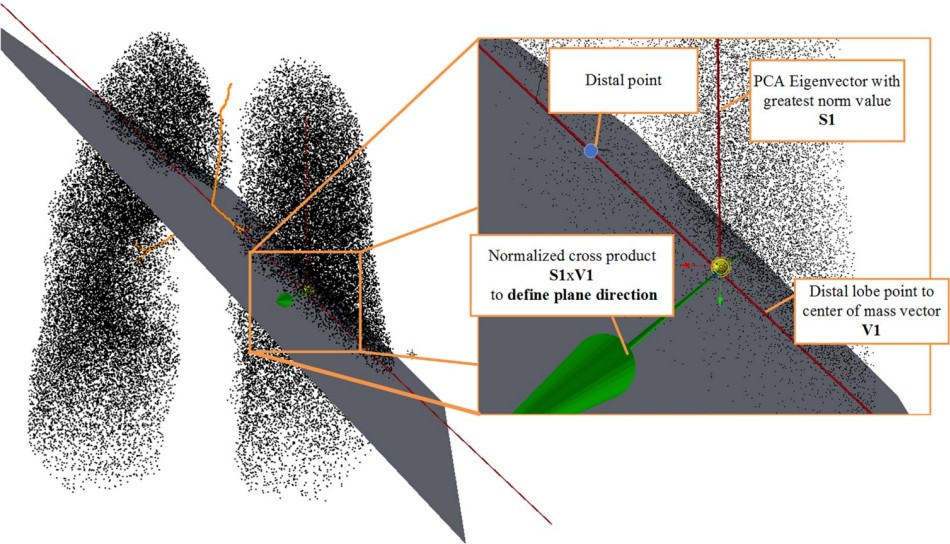

**Fig 4. Definition of splitting plane for bifurcating distributive structures.**

## 2.3 Spatial probability maps of branching properties

The location of each new generation branch is calculated as explained before and provides a random sample out of all possible bronchial tree conformations. In this step of the proposed framework we produce probabilistic maps for each generation branch that provide estimates of the spatial probability to encounter a certain generation at some point of the imaging data. Such a probabilistic model allows to optimize clinical decision making by accounting for the branches' distributional uncertainty.

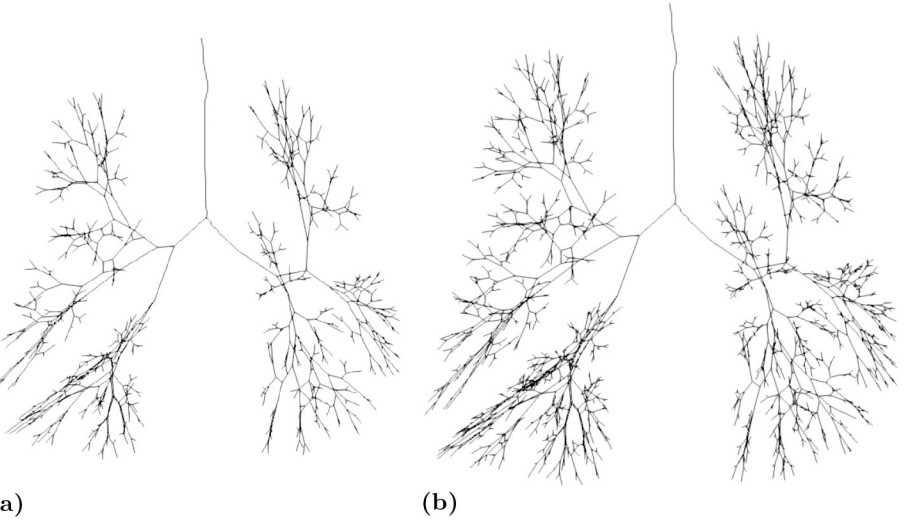

(a)                                          (b)

**Fig 5. Extended bronchial tree (a) 9 generations, (b) 12 generations.**

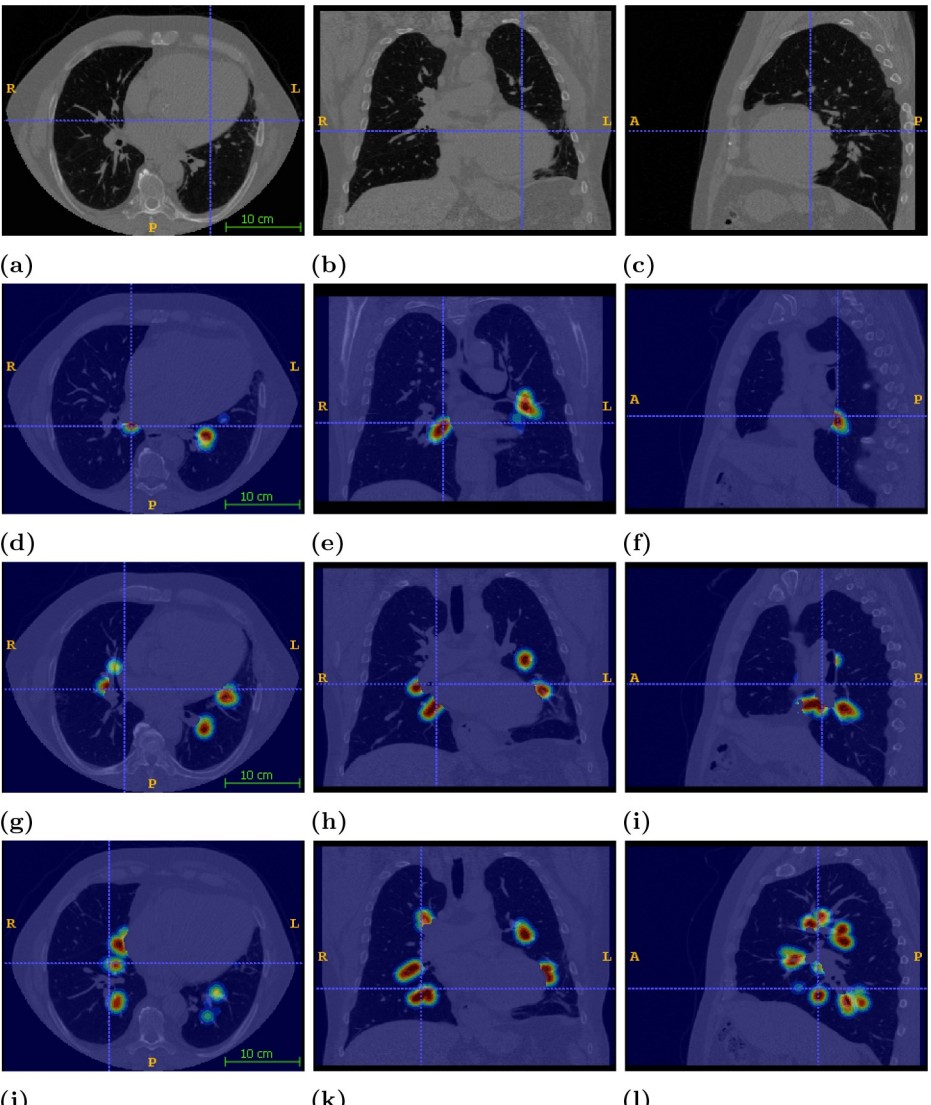

**Fig 6. Visualization of fourth generation probability map.** First column corresponds to axial view. Second column corresponds to coronal view. Third column corresponds to sagital view. The first row depicts raw imaging data, the second row presents the probabilistic maps for second generation, the third row presents the probabilistic maps for second generation and the fourth row presents the probabilistic maps for second generation.

Let's denote with $W_g(\mathbf{x})$, $I : \Omega \to \mathcal{R}$, the probability map for generation $g$ where $\mathbf{x} = (x, y, z)$, $\mathbf{x} \in \Omega$ is a voxel in the spatial domain $\Omega \subset \mathcal{R}^3$ of the volumetric imaging data. Then

$$W_g(\mathbf{x}) = \frac{1}{\sigma\sqrt{2\pi}}e^{-(d)^2/2\sigma^2} \qquad (2)$$

where $d$ is the distance of voxel $\mathbf{x}$ to the closest edge of $\mathcal{G}$ labeled with generation $g$. Parameter $\sigma$ is set experimentally to $\sigma = 1$. The extracted spatial map is overlaid on the CT scans, as shown in Fig 6, providing insightful visualization of spatial likelihood for each branching generation.

## 2.4 Surface generation on predicted 1-dimensional representation

The extracted 3D geometries are required to conduct studies on computational fluid dynamics, particle transfer and deposition, ventilation, stress analysis and deformation simulations. Marching cubes algorithm [73] is a very well established method implemented in FAST [32] allowing the generation of 3D geometric models from airway segmentation label maps. The constriction simulation method aims to generate 3D tubular surface structures with smaller diameters. To this end, Laplacian surface contraction offers a solution that deforms the geometry pushing the vertices towards the direction of the inward normals.

The extension of the extracted centerline generates a predictive representation of the bronchial tree given the available space. However, for the outcome of space-filling algorithms to be useful in fluid dynamics simulation, particle deposition simulation or stress finite element analysis based studies, the boundary conditions in the form of triangular 3D meshes need to be defined. Initially, as a simplified approach, to define the diameter of each generation we can employ the power law consistent with Murray's law of symmetric branching [33, 34].

$$d_z = d_0 \times 2^{-z/3} \tag{3}$$

where $d_0$ denotes the branch diameter of the trachea and $d_z$ the branch diameter for generation $z$.

Furthermore, if we take into account that each branch demonstrates different branching angle and diameter properties, the relation between airway diameter ($d$) and branching angles ($\theta$) is based on the following rules validated by Kamiya et al. [37] and Kitaoka et al. [2]:

$$d_0^2 = d_1^2 + d_2^2 \tag{4}$$

$$\frac{d_0^2}{\sin(\theta_1 + \theta_2)} = \frac{d_1^2}{\sin\theta_1} = \frac{d_2^2}{\sin\theta_2} \tag{5}$$

where the index 0 stands for the parent branch, and the indices 1 and 2 for the two children branches, respectively.

To reconstruct the lung surface we employ a point cloud sampling approach as input for Poisson surface reconstruction. The outcome is a smooth surface with smooth transitions instead of abrupt transitions in the intersection with the original tubular meshes. The tubular-shaped point cloud is sampled using a uniform random distribution. A clean-up step, visualized in Fig 7, ensures that no point can be found in distance less than the prescribed diameter of every available branch. The resulting point cloud is used to compute normals. A bilateral normal smoothing [74] function prepares the point cloud for Poisson surface reconstruction [75]. smoothing the point normals. This step facilitates the surface reconstruction in bifurcations and transitional parts. Furthermore, since the directed graph is extracted where each point on the centerline corresponds to a point on the lung surface it is possible to further deform the surface with a custom function or pattern. The generated surface for seven and ten generations is presented in Fig 8.

## 2.5 Simulation of constrictive pulmonary diseases' effect on airway tree

This section aims to provide the methodology for simulation of broncho-constriction allowing to subsequently estimate the dynamic behaviour of the lung airways in the case of an exacerbation event. A bronchial tree 3D geometry is the input for this process yielding as output contracted airways. The proposed geometry contraction procedure is presented by Nousias et al. [45] and Lalas et al. [46] and is an extension of the work of Au et al. [76] facilitating a Laplacian

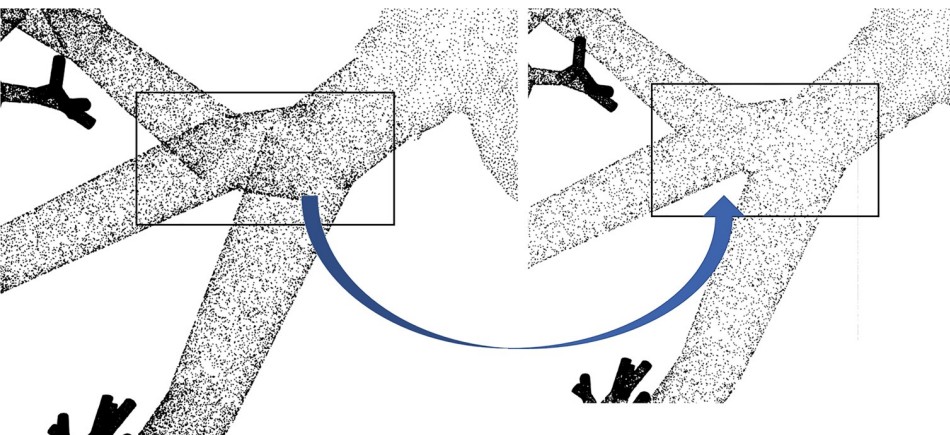

**Fig 7. Outcome of point cloud generation process.** Extra refinements remove the inner points facilitating normal estimation for Poisson surface reconstruction.

smoothing process that shifts vertices along the estimated curvature normal direction. The airway geometric model consists of connected triangles forming the boundary conditions. Each triangular mesh $\mathcal{M}$ can be described as $\mathcal{M} = \{\mathcal{V}, \mathcal{E}, \mathcal{F}\}$ where $\mathcal{V}$ is the set of vertices, $\mathcal{E}$ is the set of edges and $\mathcal{F}$ is the set of faces constituting the 3D surface. Each vertex $i$ can be represented as a point $\mathbf{v}_i = (x_i, y_i, z_i)$, $\forall i = 1, 2, \cdots, N$. For each face $\mathbf{f}_i$, $\forall i = 1, 2, \cdots, l$ we denote the centroid

$$\mathbf{m}_i = \frac{\mathbf{v}_{i_1} + \mathbf{v}_{i_2} + \mathbf{v}_{i_3}}{3}, \forall i = 1, 2, \cdots, l \tag{6}$$

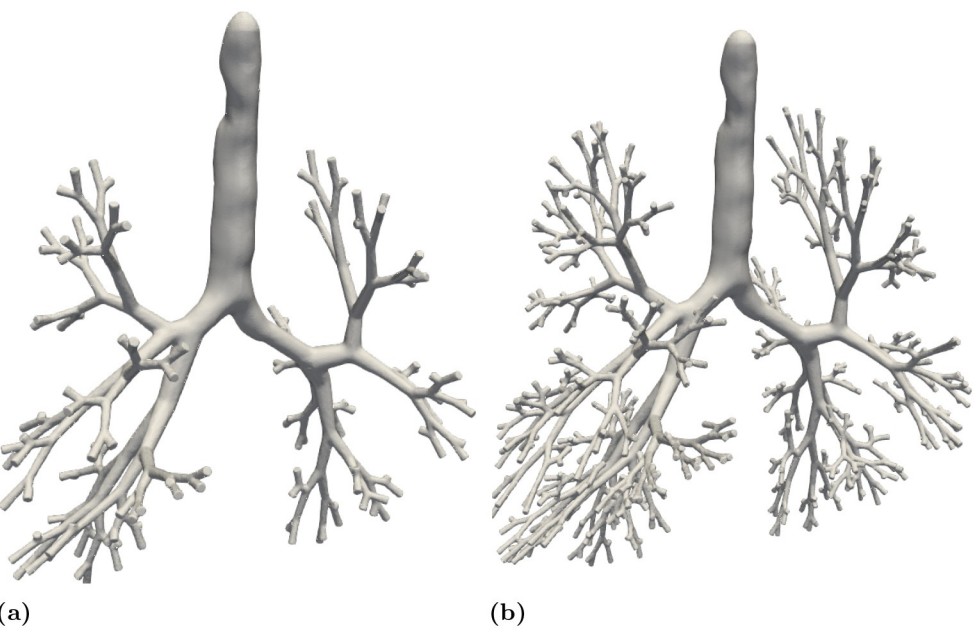

(a)  (b)

**Fig 8. Generation of surface from the extended bronchial tree centerline.**

The outward unit normal $\mathbf{n}_{m_i}$ to the face $\mathbf{f}_i$ (located at the centroid $\mathbf{m}_i$) is calculated as $\mathbf{n}_{m_i}$:

$$\mathbf{n}_{m_i} = \frac{(\mathbf{v}_{i_2} - \mathbf{v}_{i_1}) \times (\mathbf{v}_{i_3} - \mathbf{v}_{i_1})}{\| (\mathbf{v}_{i_2} - \mathbf{v}_{i_1}) \times (\mathbf{v}_{i_3} - \mathbf{v}_{i_1}) \|}, \forall i = 1, \cdots, l \tag{7}$$

where $\mathbf{v}_{i_1}, \mathbf{v}_{i_2}, \mathbf{v}_{i_3}$ are the vertices corresponding to face $\mathbf{f}_i$. Given $\mathbf{L} \in \mathcal{R}^{N \times N}$ the curvature flow Laplacian operator, the product $\delta = \mathbf{LV}$ approximates the inward curvature flow normals [77]. The motivation for employing the curvature flow Laplacian operator [78] on the mesh is that its output is not affected by mesh density. Specifically,

$$\delta = \mathbf{LV} = [\delta_1^T, \delta_2^T, \ldots, \delta_N^T]^T, \delta_i = -4A_i \kappa_i \mathbf{n}_i \tag{8}$$

where $A_i$ is the one-ring area, $\kappa_i$ is the local curvature and $\mathbf{n}_i$ is the inward curvature flow normal of the $i^{th}$ vertex.

The positions of the vertices satisfying $\mathbf{LV} = 0$ result in a zero volume string-like mesh and can be used to simulate mesh contraction. However, since such an optimization problem has more than one solutions, further constraints are required [76]. Introducing weighting matrices $\mathbf{W}_H \in \mathcal{R}^{N \times N}$ $\mathbf{W}_L \in \mathcal{R}^{N \times N}$ can smoothly move vertex positions $\mathbf{V} \in \mathcal{R}^{3 \times n}$ towards the direction of the inward unit normal by iteratively solving the following minimization problem

$$\hat{\mathbf{V}} = \arg \min_{\mathbf{V}} \{\| \mathbf{W}_L \mathbf{LV} \|^2 + \mathbf{W}_H \| \mathbf{V} - \mathbf{V}_a \|\} \tag{9}$$

where $\mathbf{V}_a \in \mathcal{R}^{3 \times N}$ corresponds to the vertex positions before the contraction at each iteration.

The weighting matrices $\mathbf{W}_H$ and $\mathbf{W}_L$ regulate the mesh contraction and mesh attraction, respectively. Initially, we set them to $\mathbf{W}_L = k \cdot \sqrt{A} \cdot \mathbf{I}$ and $\mathbf{W}_H = \mathbf{I}$, where $\mathbf{I} \in \mathcal{R}^{N \times N}$ is the identity matrix, $k$ a double constant experimentally tuned to $10^{-3}$ and $A$ the average face area of the model.

Eq (9) can be expressed as

$$\begin{bmatrix} \mathbf{W}_L \mathbf{L} \\ \mathbf{W}_H \end{bmatrix} \cdot \mathbf{V}' = \begin{bmatrix} \mathbf{0} \\ \mathbf{W}_H \mathbf{V} \end{bmatrix} \tag{10}$$

The analytical solution can be formulated as

$$\mathbf{V}' = (\mathbf{A}T\mathbf{A})^{-1} \mathbf{A}\mathbf{b} \tag{11}$$

where matrices $\mathbf{A}$ and $\mathbf{b}$ are given by

$$\mathbf{A} = \begin{bmatrix} \mathbf{W}_L \mathbf{L} \\ \mathbf{W}_H \end{bmatrix}, \mathbf{b} = \begin{bmatrix} \mathbf{0} \\ \mathbf{W}_H \mathbf{V} \end{bmatrix} \tag{12}$$

After each iteration $t$ we update the contraction and inflation weights to be used in iteration $t + 1$ so that $\mathbf{W}_L^{t+1} = s_L \mathbf{W}_L^t$ and $W_{H,i}^t = W_{H,i}^0 \sqrt{\frac{A_{0_i}}{A_{t_i}}}$, where $A_{0_i}$ and $A_{t_i}$ are the original and the current one-ring area respectively. The Laplacian matrix for iteration $t + 1$, $\mathbf{L}^{t+1}$ is also recomputed. On these grounds, to simulate broncho-constriction we require to reduce the airway diameter to a predefined level of narrowing is reached. This level is defined by certain termination criteria [45, 46]. Thus, a metric is required that measures the diameter of the bronchi under process. To estimate the degree of contraction of the airway's geometry after each iteration, we employ a shape diameter function (SDF) based scheme [79] implemented in [80] that evaluates the local volume based on the estimated local diameter assigned to each face of the

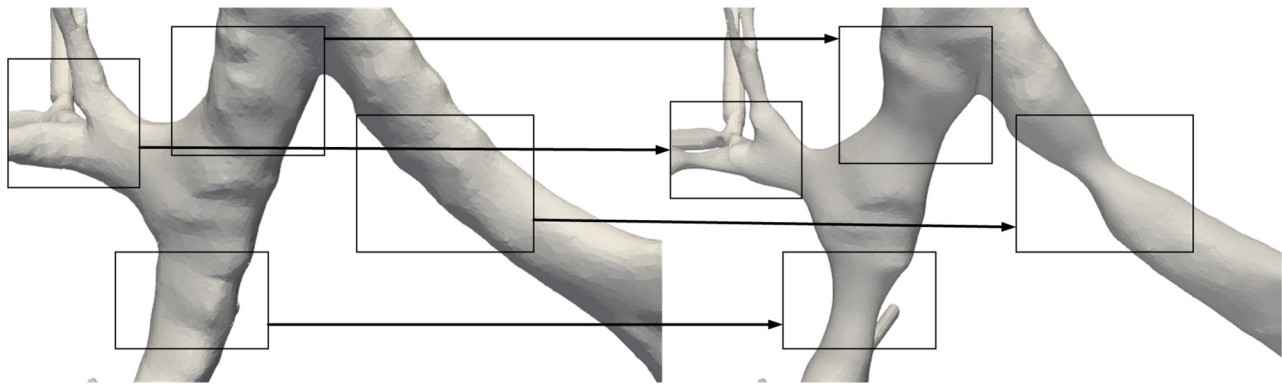

**Fig 9. Simulation of constrictive pulmonary conditions.**

mesh, also known as raw SDF values. Measuring the volume before and after the Laplacian contraction iteration can set the termination criteria. Fig 9 presents a simulation of constrictive pulmonary conditions.

## 3 Results

### 3.1 Dataset description

For the evaluation of the aforementioned approaches we employed the dataset provided by VESSEL12 (VESsel SEgmentation in the Lung) challenge [81] and EXACT09 [17]. The VESSEL dataset is comprised of 20 anonymized scans in Meta (MHD/raw) format. The latter consists of 75 completely anonymized chest CT scans contributed by eight different institutions, acquired with several different CT scanner brands and models, using a variety of scanning protocols and reconstruction parameters. The conditions of the scanned subjects varied widely, ranging from healthy volunteers to patients showing severe abnormalities in the airways or lung parenchyma. Fig 6a to 6c present imaging instances of CT slices across the axial, coronal and sagittal planes. The generation of the initial airway surface, lung volume and 1-dimensional representation are performed using the FAST framework [32].

### 3.2 Structural modelling and validation

Our simulation framework processes the initial tree centerline and generates a structural estimation given the first three to four available generation and their morphometric characteristics i.e., lengths and diameters. To facilitate the comparison with morphometric data, we employed a publicly available dataset provided by Montesantos et al. [44] labelled as *pone.0168026.s001*. For the sake of self-completeness, the authors of [44] provided morphometric data extracted from HRCT images acquired at the University Hospital Southampton NHS Foundation Trust as a part of study described in [82, 83]. Data from seven healthy subjects and six patients with moderate or persistent asthma were included in the dataset. Asthmatic patients patients were diagnosed exacerbation-free for at least one month and were male non-smokers.

A Sensation 64 slice HRCT scanner (Siemens, Enlargen, Germany) was utilized to capture 3D images from mouth to the base of the lungs. Subjects were posed in supine position and were instructed to perform slow exhalation. Groundtruth data for the development of bronchial tree models in [44] were extracted by Pulmonary Workstation 2 Software including 3 to 4 generations in the upper lobes and 6 to 7 generations in the lower lobes. For each subject, a

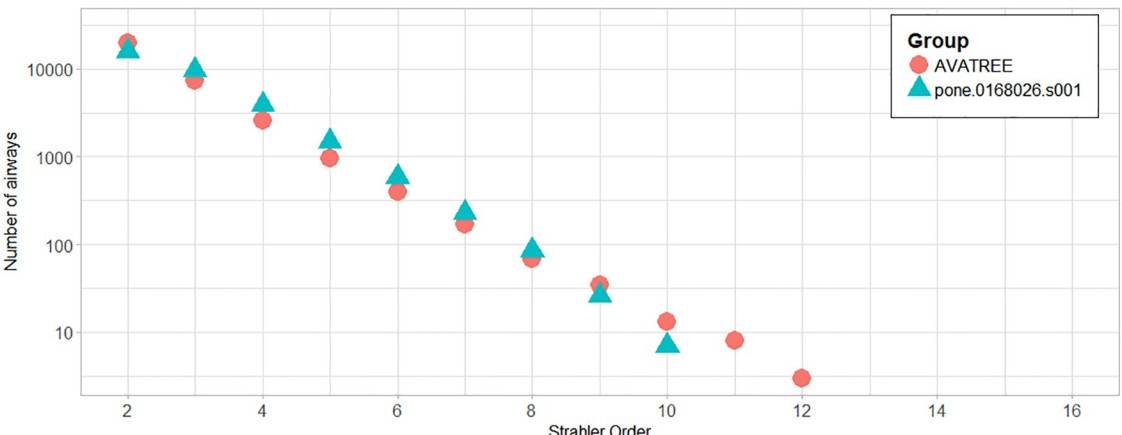

**Fig 10. Number of airways for each Strahler order for our model, "AVATREE", and "pone.0168026.s001".**

morphology file includes the total lung volume of the subject lung (in $cm^3$) and the percent volume per lobe while a translation file contains the airway connectivity, starting from the trachea to the terminal nodes. We used the generated trees from [44] to validate our approach and compare our results with relevant literature findings. Specifically, we compared the distributions of diameters, airway lengths and branching angles for each generation and the total number of airways for Horsfield and Strahler orders.

In total 31204 acini were calculated being in agreement with the results reported by [6, 44]. Figs 10 and 11 present a comparison in terms of the number of airways for each level of Strahler and Horsfield orders. This comparison confirms that our model comes into agreement with *pone.0168026.s001*. Furthermore, distributions of airway lengths, branching angles and diameters were plotted for each generation, for AVATREE and *pone.0168026.s001* [44]. Airway lengths maintain the same exponential decay pattern for both models. Differences appear in generations 1 to 4 that are distinctively defined by body size and anatomical features. The distribution of branching angles of our model is also confirmed by *pone.0168026.s001* [44] maintaining a nearly linear decay with a very small rate. The distributions of diameters per generation are also observed to follow an exponential decay pattern. Both our model and

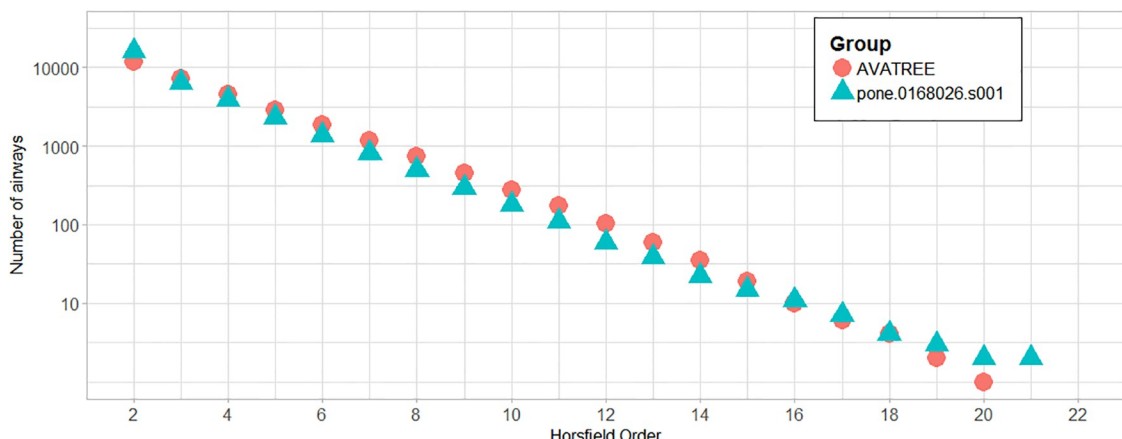

**Fig 11. Number of airways for each Horsfield order for our model, "AVATREE", and "pone.0168026.s001".**

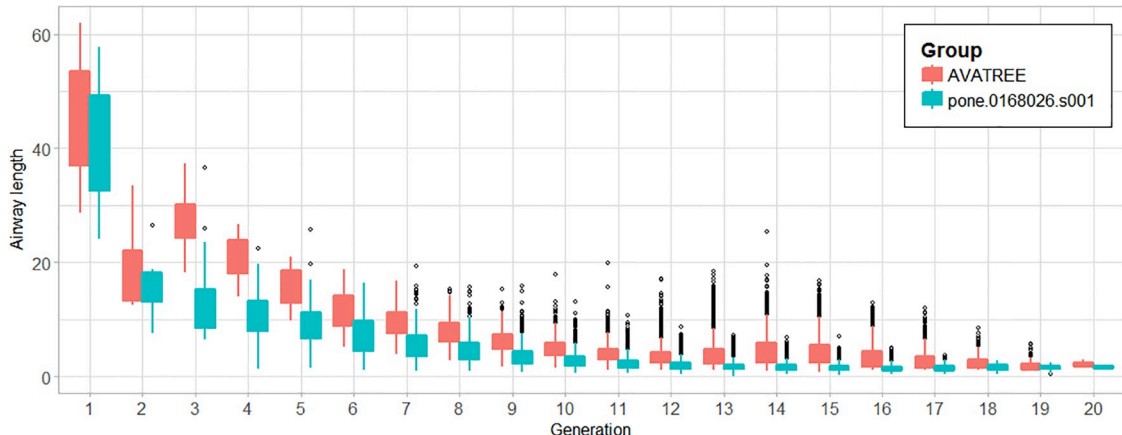

**Fig 12. Distribution of airway lengths for each generation for AVATREE and *pone.0168026.s001*.**

*pone.0168026.s001* [44] decay similarly after generation 4 validating the morphometric characteristics of the airway trees generated by our approach. Figs 12 to 14 present the distribution of airway length, branching angle and diameter for each generation for AVATREE and for *pone.0168026.s001* [44]. Table 1 presents and overview of quantitative macroscopic figures for AVATREE and relevant studies. Branching ratios ($RB_H$, $RB_S$), diameter ratios ($RD_H$, $RD_S$) and length ratios $RL_H$, $RL_S$) were calculated for Strahler and Horsfield ratios denoted as $^*_S$ and $^*_H$ respectively. Specifically, $RB_H$, $RD_H$ and $RL_H$ were calculated equal to $RB_H = 1.74$, $RD_H = 1.259$ and $RL_H = 1.26\pm1.01$. Montesantos et al. [44] reported $RB_H = 1.56$, $RD_H = 1.115$ and $RL_H = 1.13$ respectively. Additionally, $RB_S$, $RD_S$ and $RL_S$ were calculated equal to $RB_S = 2.35$, $RD_S = 1.25$ and $RL_S = 1.23\pm1.02$ and are close to the figures provides by relative studies [1, 44] as Table 1 reveals. Likewise, rate of decline for diameters per generation $RD$ was calculated to $RD = 0.83\pm0.21$, being in agreement to [44]. Average branching angle $\theta$ for our model was calculated to $32.44\pm28.95$ comparable to [44] reporting a $\theta$ equal to $42.1\pm21.4$. Finally, Figs 15 and 16 present bronchial tree 1-dimensional representations extended up to 12 and 23 generations respectively. Additionally, for both generated models the surface has been reconstructed for the first 7 generations.

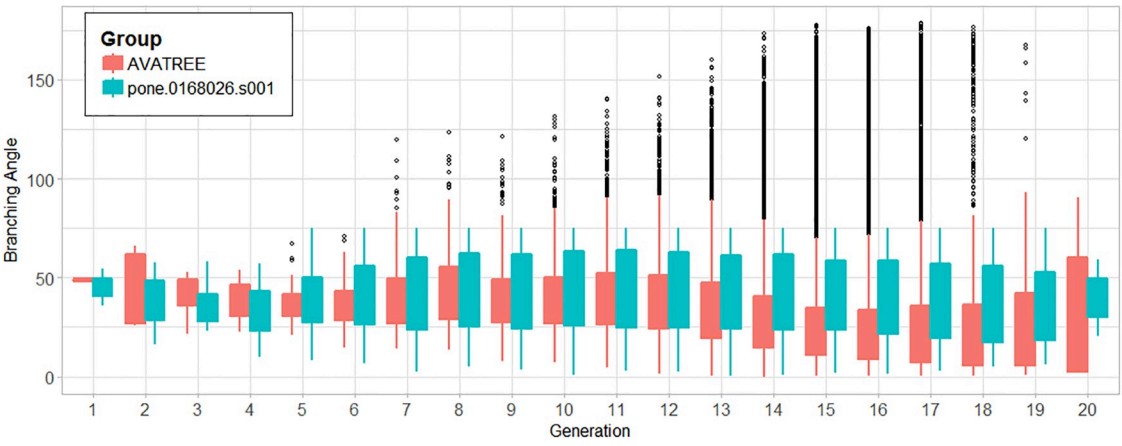

**Fig 13. Distribution of branching angles for each generation for AVATREE and *pone.0168026.s001*.**

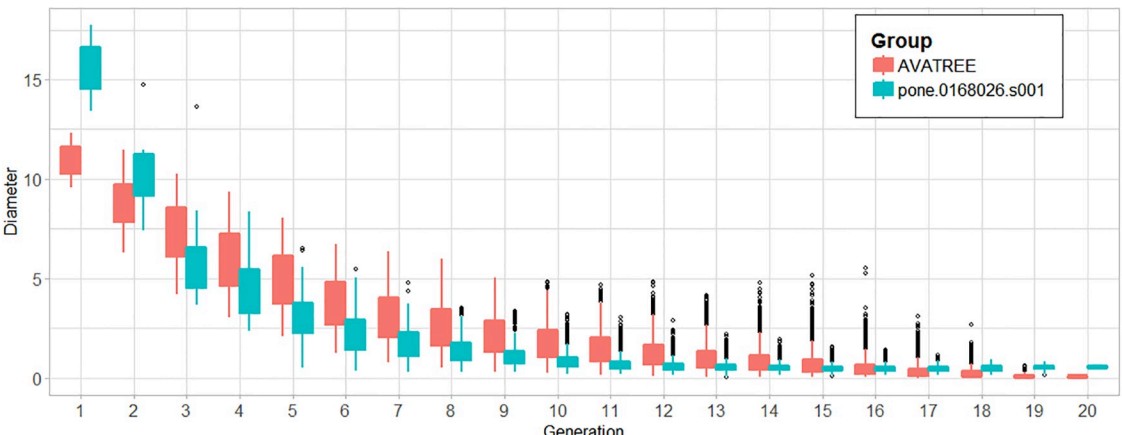

**Fig 14. Distribution of diameters for each generation for AVATREE and *pone.0168026.s001*.**

## 3.3 Open-source library & Graphical User Interface

The presented components of AVATREE are provided as an open-source solution publicly available at (https://gitlab.com/LungModelling/avatree) accompanied by a graphical user interface (GUI). The implemented application programming interface (API) includes the following modules, a) input-output functionalities, b) 1-dimensional representation tools including centerline extraction, graph generation, derivation of graph node properties c) bronchial tree extension tools extending the 1-dimensional representation to the desired number of generations, d) 3D surface generation and processing tools and e) airway narrowing simulation tools.

The GUI, presented in Fig 17, employs the set of functionalities defined by AVATREE and is comprised of four panels, namely data input and output panel, area selection panel, segmentation panel and broncho-constriction simulation panel. Through the GUI the user can load a 3D model, select the area to be processed, as Fig 18 visualizes, and use the narrowing functionalities to reduce the airway diameter by the desired percentage. The amount of narrowing depends on the number of iterations and contraction weight multiplier. In Fig 18 an airway of the first generation was constricted by 66%. The deformed surface introduced into computational fluid dynamics can provide insight into the breathing pattern and drug delivery in asthmatic lungs [84]. In the segmentation panel, the surface faces can be classified based on local properties. The one illustrates the shape diameter function (SDF) [79], while the other one the 3D surface according to the generation number. The results are visualized in Fig 19.

**Table 1. Structural features comparison.**

|  | No of acini | Diameter Rate of decline | $RB_H$ | $RD_H$ | $RL_H$ | $RB_S$ | $RD_S$ | $RL_S$ | Mean $\theta$ |
|---|---|---|---|---|---|---|---|---|---|
| **AVATREE** | 31204 | 0.83±0.21 | 1.74 | 1.259 | 1.26±1.01 | 2.35 | 1.25 | 1.23±1.02 | 32.4488±28.95 |
| **Tawhai et al**. [6] | 29445 | | 1.47 | | 0.13 | 2.8 | 1.41 | 1.39 | |
| **Horsfield et al**. [1] | 27992 | | | | | 2.54-2.81 | 1.5 | 1.55 | 37.28 |
| **Bordas et al**. [7] | | | | | | | | | 42.90±0.10 |
| **Montesantos et al**. [44] | 27763±7118.5 | 0.789±0.16 | 1.56 | 1.116 | 1.13 | 2.49 | 1.397 | 1.392 | 42.1±21.4 |

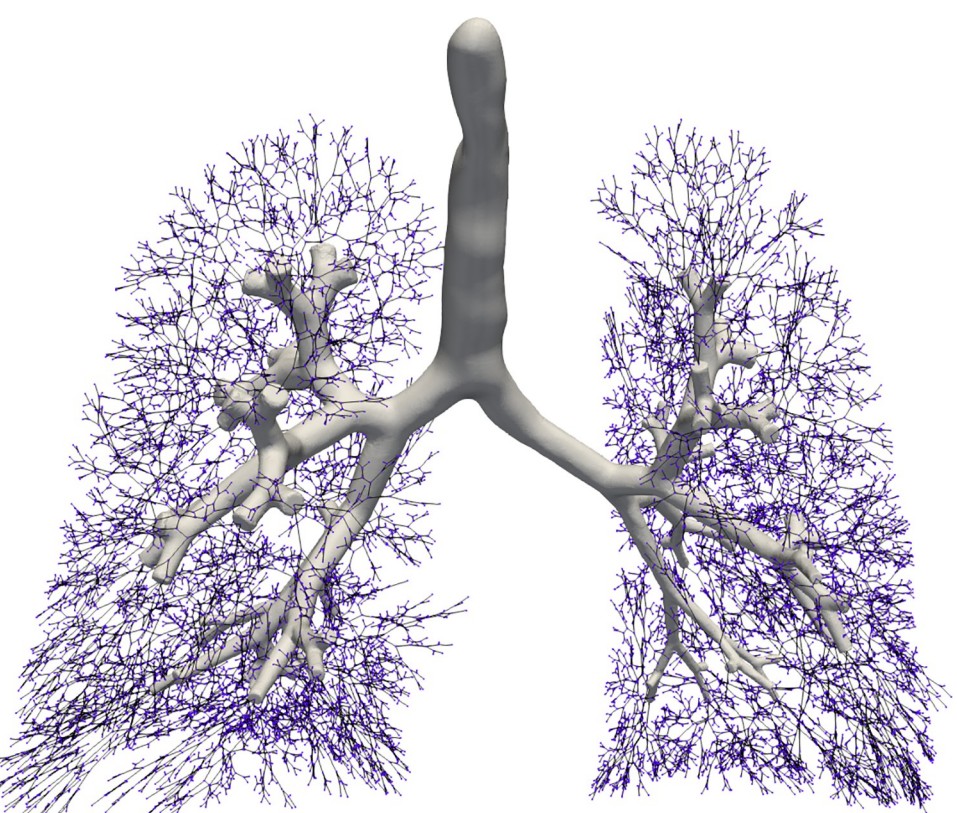

**Fig 15. Estimation of bronchial tree for 12 generations.** Surface reconstruction was performed for the first 7 generations.

## 4 Conclusion

In this paper, we presented AVATREE, an end-to-end approach modeling the subject-specific airway tree that defines the personalized boundary conditions required for the simulation of pulmonary function. This particular personalization aspect refers to the extraction of the main airways and the lung volumes from imaging allowing the simulation of a personalized extended bronchial tree. The utter goal in this category of studies is to eventually predict gas flow and particle distribution in healthy and constricted bronchial trees. Modelling lung ventilation patterns can provide grounds for performance and fatigue estimation in high-frequency ventilation cases and give insight into drug delivery or even transfer of harmful particulates. Specifically, this work presents an open-source simulation framework that utilizes imaging data to provide patient-specific representations of the structural models of the bronchial tree. The extended 1-dimensional centerline facilitates the generic estimation of pulmonary function through lumped 0-dimensional studies and allows the generation of probabilistic confidence maps of airway generation data. Such a visualization could be exploited by airway tree segmentation methodologies to improve the results further constraining the 3D space to be searched. Further elaborating on the benefits of the presented methodology, the generation of extended bronchial tree surface allows the assessment of airway functionality. Several studies available in the literature have employed computational fluid dynamics to predict flow and particle transfer patterns inside the conducting regions of the bronchial tree. Generating the 3D mesh constituting the surface defines the boundary conditions for this category of studies.

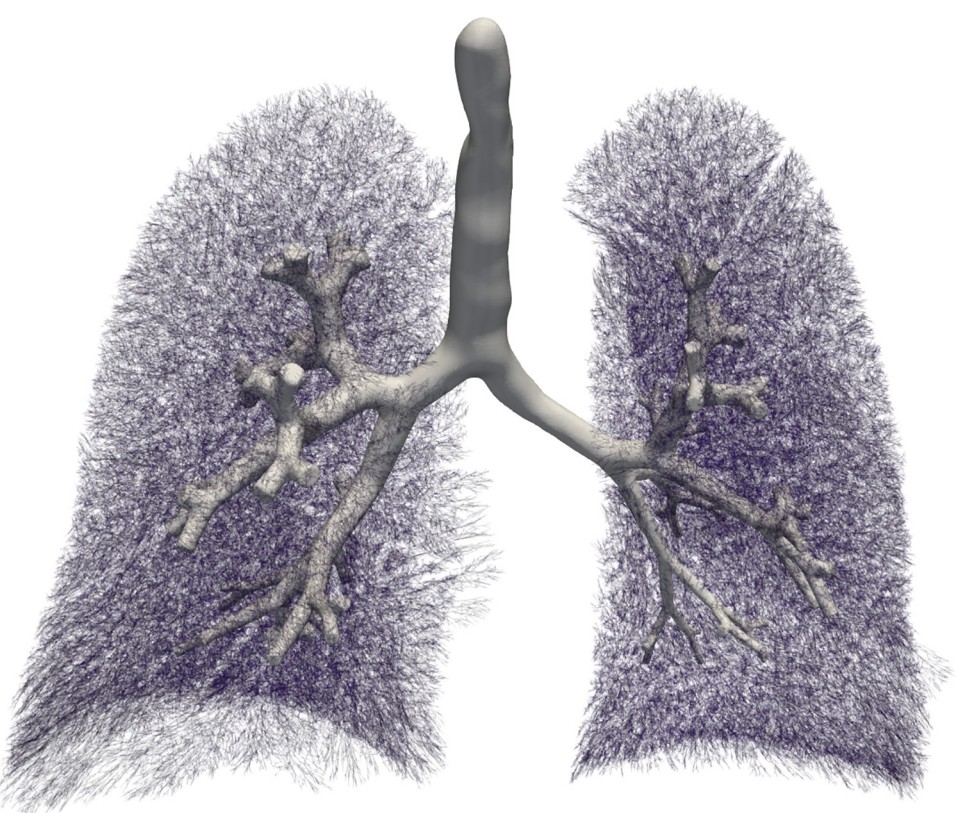

**Fig 16. Estimation of bronchial tree for 23 generations.** Surface reconstruction was performed for the first 7 generations.

Furthermore, surface deformation functionalities allow simulating broncho-constriction, which is the main feature in constrictive pulmonary diseases such as asthma and COPD. Existing approaches of medication adherence in asthma and COPD patients are usually based on the analysis of breathing signals acquired with acoustic sensors attached on inhaler devices

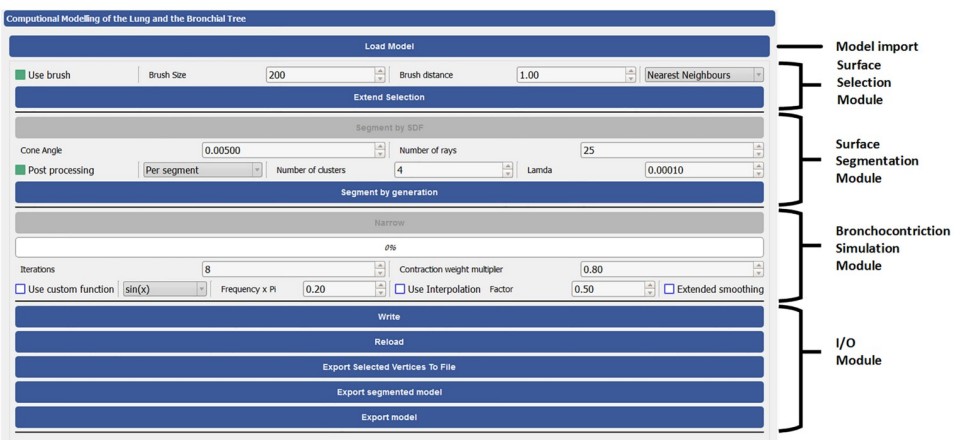

**Fig 17. User interface.**

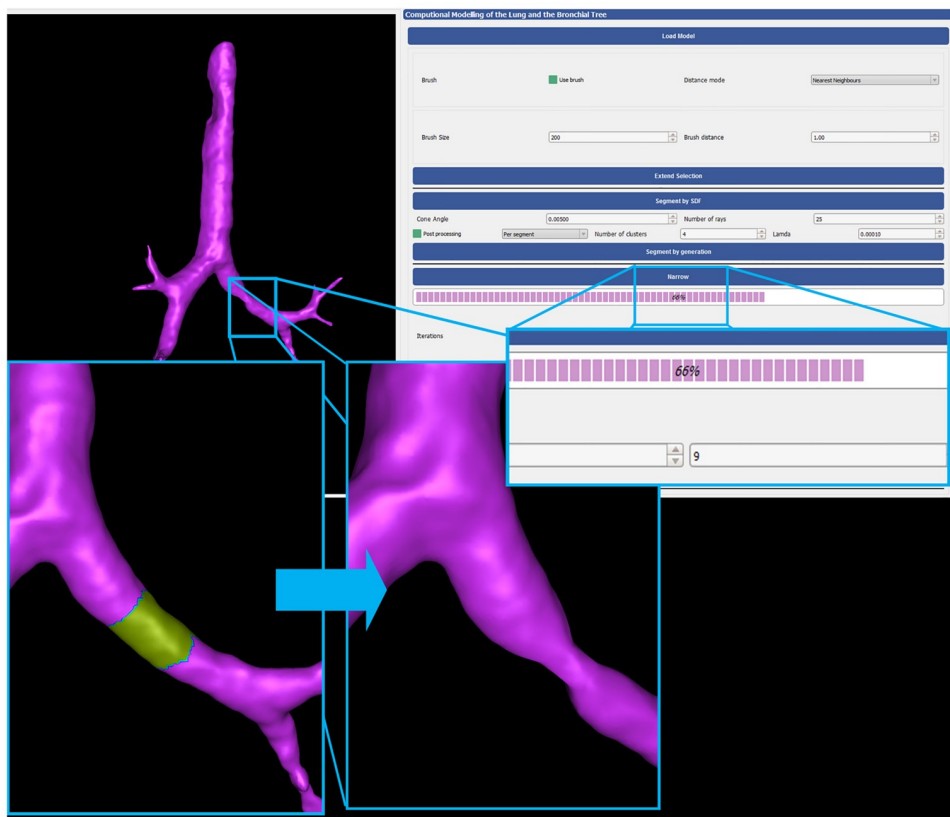

**Fig 18. Broncho-constriction simulation.** Airway of second generation narrowed at 34% of original diameter.

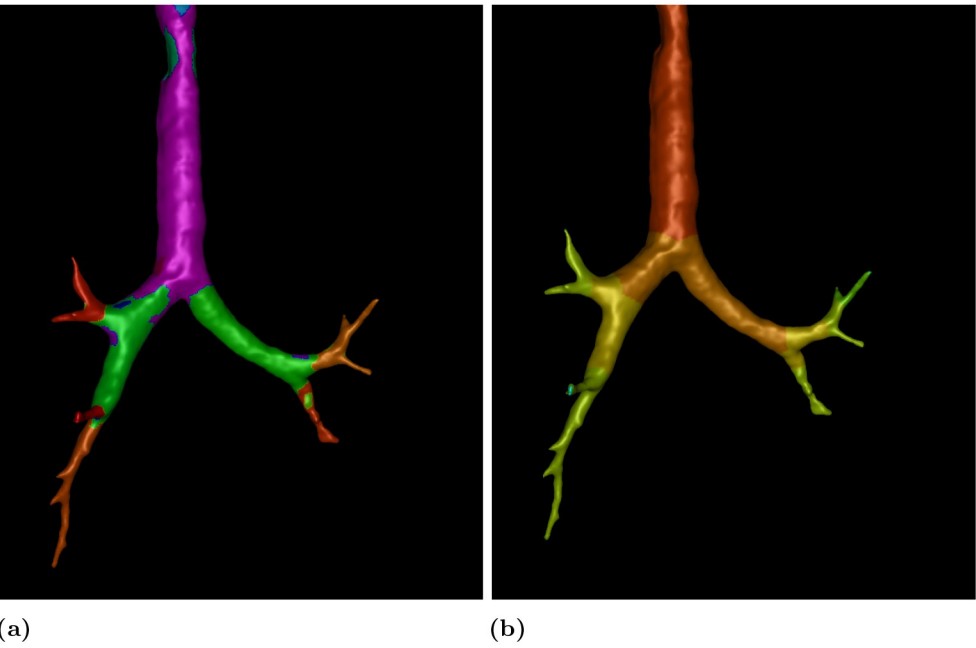

(a)                                                    (b)

**Fig 19. Surface annotation with a) SDF function visualizing local diameter b) airway generation.**

[85] [86]. The concept of such studies is to facilitate self-management by guiding the patients to improve their inhaler usage technique [87]. AVATREE could contribute to this type of analysis by estimating the effectiveness of inhaled medication based on personalized imaging data and particle deposition simulations [46]. Both automated and UI guided solutions are provided by the presented open-source solution enabling users to simulate pathological conditions in asthmatic patients guided by imaging priors data from healthy subjects.

## Author Contributions

**Conceptualization:** Stavros Nousias, Evangelia I. Zacharaki.

**Formal analysis:** Stavros Nousias, Evangelia I. Zacharaki.

**Investigation:** Stavros Nousias.

**Project administration:** Konstantinos Moustakas.

**Software:** Stavros Nousias.

**Supervision:** Evangelia I. Zacharaki, Konstantinos Moustakas.

**Visualization:** Stavros Nousias.

**Writing – original draft:** Stavros Nousias.

**Writing – review & editing:** Evangelia I. Zacharaki, Konstantinos Moustakas.

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
