## [Decision Letter · Decision Letter 0]

7 Oct 2019

PONE-D-19-24923

AVATREE: An open-source computational modelling framework modelling Anatomically Valid Airway TREE conformations

PLOS ONE

Dear Mr Nousias,

Thank you for submitting your manuscript to PLOS ONE. After careful consideration, we feel that it has merit but does not fully meet PLOS ONE’s publication criteria as it currently stands. Therefore, we invite you to submit a revised version of the manuscript that addresses the points raised during the review process.

We would appreciate receiving your revised manuscript by Nov 21 2019 11:59PM. To enhance the reproducibility of your results, we recommend that if applicable you deposit your laboratory protocols in protocols.io, where a protocol can be assigned its own identifier (DOI) such that it can be cited independently in the future. For instructions see: http://journals.plos.org/plosone/s/submission-guidelines#loc-laboratory-protocols

We look forward to receiving your revised manuscript.

Kind regards,

Fang-Bao Tian

Academic Editor

PLOS ONE

Journal Requirements:

'This research has been funded by the European Regional Development Fund of the

European Union and Greek national funds through the Operational Program Compet-itiveness, Entrepreneurship and Innovation, under the call RESEARCH - CREATE - INNOVATE (project code:T1EDK-03832).'

'The author(s) received no specific funding for this work.'

3.  Please ensure that you refer to Figure 7 in your text as, if accepted, production will need this reference to link the reader to the figure.

Additional Editor Comments (if provided):

Thank you for submitting your work to Plos One. It has been commented by two independent specialists. As you can see, their views regarding the paper quality are quite different. I would give you an opportunity, to clarify the major issues raised by Reviewer 2, given that the tool introduced may have important application in the community.

Reviewers' comments:

Reviewer's Responses to Questions

**Comments to the Author**

1. Is the manuscript technically sound, and do the data support the conclusions?

Reviewer #1: Yes

Reviewer #2: Partly

2. Has the statistical analysis been performed appropriately and rigorously? 

Reviewer #1: Yes

Reviewer #2: No

3. Have the authors made all data underlying the findings in their manuscript fully available?

Reviewer #1: Yes

Reviewer #2: No

4. Is the manuscript presented in an intelligible fashion and written in standard English?

Reviewer #1: Yes

Reviewer #2: No

5. Review Comments to the Author

Reviewer #1: I have read PONE-D-19-24923 which develops a computational modelling framework called “AVATREE” to define the personalized boundary conditions required for the simulation of pulmonary functions. Specifically, the AVATREE works as follows: (i) using image data to provide patient-specific representations of the structural models of the bronchial tree, (ii) establishing and extending 1D graph representations of the bronchial tree, (iii) generating 3D surface models of the extended bronchial tree models, (iv) producing probabilities visualization of airway generations on the personalized CT imaging data, and (v) offering an open-source toolbox in C++ and a graphical user interface integrating modelling functionalities. The AVATREE can successfully create anatomically valid airway tree conformations, which would be helpful to eventually predict gas flow and particle deposition characteristics in the healthy and diseased bronchial trees. In my view, this computational modelling framework is worth popularizing, and therefore I advise the paper to be accepted by the PLOS ONE.

Reviewer #2: This article presents AVAtree, a software designed to take CT images and create computational descriptions of the pulmonary airways. The aim of the paper is to provide an end to end tool that creates airway trees from imaging, which would be highly useful for the community. However, there are several issues with the presentation of the models employed in the context of existing studies and softwares in the area, and there is not enough information on whether the end-point models of the airway anatomy reflect real lung structures to thoroughly assess the validity of the methods.

1. It was not clear from the manuscript how to access or use AVAtree, as this is a proposed opensource software. The link to a gitlab repository is given as a foot note, but clearer instructions would be beneficial. An attempt to access this repository at https://gitlab.com/vvr/LungModelling led to a ‘page not found’ error. Therefore it was impossible to assess whether the software behaves as described.

2. There appears to be some confusion regarding the existing literature in the background section. This section is somewhat unfocused, and needs work to clearly motivate the research being conducted. For example, Fernandez et al. looked at generating lung surface representations but this is the only study looking at the lung surface in this section. It appears to be presented in respect to defining boundary conditions for fluid dynamics models, but as far as I am aware the Fernandez model has not been used in that way. The section then jumps to a Hegedus who look at idealised representations of the upper airways, which is not related to the Fernadez study.

3. Reference [4] is incorrectly given as a reference for AVAtree on page 2.

4. Section 1.1.3 – I think the citation for coupled 3D-1D models is incorrect. Tawhai & Lin [37] is a review article, and to state that this study did the modelling is not correct – the original papers are cited in that reference.

5. The Florens model is not defined very well in section 1.1.3 and its relation to the two Tawhai studies that surround it is not clear.

6. Again, Tawhai [10] is a review, it is not the paper that defined the volume filling branching model, and it did not focus on constrictive lung conditions. The Bordas study is one of many studies that use, or build upon these methods to predict function (see below), and the other studies are neglected from discussion. The Varner paper in this paragraph seems to me to refer specifically to branching mophogenesis in lung development, which is different to representing developed morphology in a model.

7. A similar open-source pipeline is available in the Chaste framework – presented in a Bordas paper cited in this manuscript. This is not really discussed and should be as the two frameworks are similar in many ways.

8. Section 2.1 (Segmentation and airways centerline extraction). This appears to be a new implementation for airway segmentation. But how do the authors know this is accurate? Several software, including opensource software have been generated to segment airways. The employed algorithm needs to be compared in some way to the field. This should be a comparison with existing algorithms and/or a comparison to a gold standard (perhaps a manual segmentation).

9. The branching method described first by Tawhai [4], was first modified by the same group, (Tawhai et al. Journal of Applied Physiology 97: 2310-2321, 2004). Variations of this algorithm have been implemented by Bordas et al [cited] who provided an open source implementation. But variations have also been presented by Abadi et al (IEEE transactions on medical imaging 37: 693-702, 2018) and Mullally et al. (Ann Biomed Eng 37: 286-300, 2009), and Montesantos et al. PloS one 11: e0168026, 2016). In general, modifications have been made to suit a groups own modelling applications – to claim these changes as an improvement, would require some improved comparison to morphological studies. For the most part these models perform fairly similarly, and are good representations of the airway structure of the lung.

10. The difference between this and previous methods is a PCA based splitting of seed points, very limited information is given about why one would do this (it seems more complicated than other methods), or what improvements it brings. The algorithm should at the very least be compared to real lung morphometric data, as has been typical in previous studies using similar algorithms.

11. It is not clear how the volume of the lung is defined from imaging, or how generated branches are restricted to lie in the lung volume (if at all).

12. Given there is presumably a complete 3D representation of the upper airway tree from CT, some information needs to be given regaring why would want to represent these branches as idealised tubular structures for CFD?

13. Figures associated with dataset collection (and some beyond) have no numbers

14. It is not at all clear what is meant by personalised boundary conditions, or how they are calculated – what is being simulated for which personalised boundary conditions are needed? Or do these boundary conditions relate to tree generation?

15. To what level is the extended tree generated? The figures suggest that this is not to the level of the acinus as in most previous studies using similar algorithms, but that this is tunable. More information on this would be beneficial, especially in relation to how well the lung is represented it different choices are made.

16. Fig 2 shows one very ling thin branch distending from the main upper airway tree – this does not look anatomical, I would suggest that the algorithm has missed some branches from this main airway, as a very long and thin branch like that is unusual.

6. PLOS authors have the option to publish the peer review history of their article (what does this mean?). If published, this will include your full peer review and any attached files.

Reviewer #1: No

Reviewer #2: No

---

## [Author Response · Author response to Decision Letter 0]

3 Dec 2019

Reviewer 1

General comments

I have read PONE-D-19-24923 which develops a computational modelling framework called \\AVATREE" to define the personalized boundary conditions required for the simulation of pulmonary functions. Specifically, the AVATREE works as follows: (i) using image data to provide patientspecific representations of the structural models of the bronchial tree, (ii) establishing and extending 1D graph representations of the bronchial tree, (iii) generating 3D surface models of the extended bronchial tree models, (iv) producing probabilities visualization of airway generations on the personalized CT imaging data, and (v) offering an open-source toolbox in C++ and a graphical user interface integrating modelling functionalities. The AVATREE can successfully create anatomically valid airway tree conformations, which would be helpful to eventually predict gas flow and particle deposition characteristics in the healthy and diseased bronchial trees. In my view, this computational modelling framework is worth popularizing, and therefore I advise the paper to be accepted by the PLOS ONE.

Response: We would like to thank the reviewer for commending our work. With this review, we aim to provide more justification for the strong and weak points of our work and improve the organization of our paper.

Reviewer 2

General comments

This article presents AVATree, a software designed to take CT images and create computational descriptions of the pulmonary airways. The paper aims to provide an end to end tool that creates airway trees from imaging, which would be highly useful for the community. However, there are several issues with the presentation of the models employed in the context of existing studies and 1software in the area, and there is not enough information on whether the end-point models of the airway anatomy reflect real lung structures to thoroughly assess the validity of the methods. XResponse: We thank the reviewer for identifying these presentation issues. We describe below how we tried to improve the context of the related work and the methodological assessment and validation. 

Comment 2.1: It was not clear from the manuscript how to access or use AVAtree, as this is a proposed opensource software. The link to a gitlab repository is given as a footnote, but clearer instructions would be beneficial. An attempt to access this repository at https://gitlab.com/vvr/LungModelling led to a ‘page not found’ error. Therefore it was impossible to assess whether the software behaves as described. 

Response 2.1: We would like to thank the reviewer for pointing out this and to clarify that the repository was not made publicly available as we were expecting the outcome of the review process.

Actual changes implemented:

• The repository is now publicly available at

https://gitlab.com/LungModelling/avatree

• Furthermore, we attach the following private link containing the outcomes of the presented

pipeline for the needs of the review process.

https://www.dropbox.com/sh/kkh8p8eikxdvkah/AACUP0-hubPx69Waaamyvh9Ta?dl=0

Please treat both links confidentially.

Comment 2.2 There appears to be some confusion regarding the existing literature in the background section. This section is somewhat unfocused and needs work to clearly motivate the research being conducted. For example, Fernandez et al. looked at generating lung surface representations but this is the only study looking at the lung surface in this section. It appears to be presented in respect to defining boundary conditions for fluid dynamics models, but as far as I am aware the Fernandez model has not been used in that way. The section then jumps to a Hegedus who look at idealised representations of the upper airways, which is not related to the Fernadez study.

Response: We would like to thank the reviewer for his/her comment that gives us the opportunity to explain better the contribution of our work and to present the main differences in comparison with similar previous work. Our study aims towards defining personalized boundary conditions that allow to derive flow and particle deposition patterns. Airway branch generation algorithms predict the structure of the bronchial tree in regions that CT based segmentation is not possible. Other studies revealing the relation of the children branch diameters as a function of the branching angle and the parent branch diameters facilitate the definition of the expected diameter for each branch. However, to allow the derivation of flow patterns, mesh structures define the boundary conditions for the CFD simulation. To this end the work of Fernandez was employed by Tawhai et al.[1, 2, 3] facilitating the definition of the lung surface cubic Hermite interpolation. Hegedus et al. attempted to define mathematically the surface of an idealized bifurcation. Such a modelling approach, given the centerline of the bronchial tree, can define the surface of the whole tree since the airways would follow a tubular structure linked with the bifurcation surface. Thus, the aforementioned studies are relevant to our approach since we define the same boundary 2conditions as the Poisson reconstructed surface of a sampled point cloud in order to avoid the definition of special rules in the reconstruction of the surface of bifurcations.

Actual changes implemented: The whole section 1.1 was restructured to better reflect the

aforementioned reasoning.

Comment 2.3

Reference [4] is incorrectly given as a reference for AVAtree on page 2.

Response: Indeed, reference [4] was incorrectly given as a reference for AVATREE.

Actual changes implemented: The reference was removed.

Comment 2.4 Section 1.1.3 I think the citation for coupled 3D-1D models is incorrect. Tawhai and Lin [37] is a review article, and to state that this study did the modelling is not correct the original papers are cited in that reference.

Response: We would like to thank the reviewer for his/her comment that gives us the opportunity to provide sharper insights on current literature. The original paper that should be referenced in this section might be the work of Lin C-L and Tawhai M-H entitled "Multiscale simulation of gas flow in subject-specific models of the human lung".

Actual changes implemented: The correct references are now included.

Comment 2.5 The Florens model is not defined very well in section 1.1.3 and its relation to the two Tawhai studies that surround it is not clear.

Response: We would like to thank the reviewer for his/her comment that gives us the opportunity to better present the "Background and related work section". The Florens model, though belonging in the generic category of studies related to the anatomical and functional model of the human tracheo-bronchial tree is not directly linked to our study.

Actual changes implemented: Taking into account the reviewer comments 2.2, 2.5, 2.6 the whole section was rewritten.

Comment 2.6 Again, Tawhai [10] is a review, it is not the paper that defined the volume filling branching model, and it did not focus on constrictive lung conditions. The Bordas study is one of many studies that use or build upon these methods to predict function (see below), and the other studies are neglected from discussion. The Varner paper in this paragraph seems to me to refer specifically to branching morphogenesis in lung development, which is different to representing developed morphology in a model.

Response: Tawhai et al in the review paper entitled "Computational modeling of the airway and pulmonary vascular structure and function: development of a "lung physiome" provided an overview of studies related to airway bronchoconstriction mechanisms which could be significant in terms of defining and describing constrictive pulmonary conditions. However, as the reviewer correctly highlights the background section requires refocusing towards the studies that relate with the benefits of our approach. To this end, the phrase "with respect to pulmonary constrictive conditions" was removed. Bordas et al. [4] developed image analysis and modelling pipeline and applied it to healthy and asthmatic patient scans to produce complete personalized airway models to the acinar level incorporating, lung and lobar segmentation, airway segmentation and centerline extraction, algorithmic generation of distal airways, zero-dimensional models. Their results and implementation were included into the Chaste framework [5] which is an open-source framework to facilitate computational modelling in heart, lung and soft tissue simulations. The Varner paper was indeed not so relevant in this context. However Varner et al. report that "detailed morphometric analysis of the bronchial tree has revealed a similar geometric scaling. Using casts of human lungs, Weibel and Gomez reported that the average diameter of the zth generation of airways, d(z), follows the scaling law d(z) = d0 × 2− z3 , which is consistent with Murray’s law for symmetric branching" which is relevant to a certain component of our pipeline. To this end, the whole section was rewritten to better highlight such points.

Actual changes implemented: To this end, the whole section was rewritten to better highlight

the reviewer’s aspect.

Comment 2.7 A similar open-source pipeline is available in the Chaste framework { presented in a Bordas paper cited in this manuscript. This is not really discussed and should be as the two frameworks are similar in many ways.

Response: Bordas et al. [4] developed an image analysis and modelling pipeline and applied it to healthy and asthmatic patient scans to produce complete personalized airway models to the acinar level incorporating lung and lobar segmentation, airway segmentation and centerline extraction, algorithmic generation of distal airways, zero-dimensional models. Their results and implementation were included into the Chaste framework [5] which is an open-source framework to facilitate computational modelling in heart, lung and soft tissue simulations. With respect to Chaste, our work: • generates surface meshes of extended patient-based bronchial trees,

• simulates constrictions of the airways,

• generates models that are suitable for computational fluid dynamics (CFD) simulations,

• generates probabilistic view of airway tree generations.

Actual changes implemented: A new paragraph was added in 1.1 discussing Chaste framework and relevant similar frameworks.

Comment 2.8 Section 2.1 (Segmentation and airways centerline extraction). This appears to be a new implementation for airway segmentation. But how do the authors know this is accurate? Several software, including opensource software, have been generated to segment airways. The employed algorithm needs to be compared in some way to the field. This should be a comparison with existing algorithms and/or a comparison to a gold standard (perhaps a manual segmentation).

Response: We would like to thank the reviewer for his/her comment that gives us the opportunity to strengthen the basis of our pipeline. The input of our approach is the binary masks of the segmented airway tree and the lung volumes and can be obtained by any algorithm that successfully segments the first generations of the airways.For this purpose we investigated two algorithms, but AVATREE is modular to the CT segmentation method. The first investigated algorithm is the gradient vector flow [6, 7] which achieved high accuracy with low false-positive rate (only 1.44%) in a comparative study [8] in the context of the EXACT09 airway segmentation challenge. However the implementation of this method showed some instabilities, therefore we also exploited a standard and stable approach, which is the seeded region growing algorithm proposed in [9]. All algorithms presented in EXACT09 study perform well in the prediction of the first 4 generations validated on images of 16 subjects. As presented by Lo et al. the following figure indicates all 15 approaches discussed succeed in detecting the first four generations highlighted with the green color. Thus, our approach receives as input the green region of the airways detected by available baseline and sophisticated algorithms assumed to be a "golden" standard" . To this end we investigated both implementations. The former can be found in the public repository located in [10] and the latter in the public repository located in [11]. An example of the method outcome is presented in Figures 1 and 2. Further justification can be provided in [8] stating that "no 5algorithm comes close to detecting the entire reference airway tree. The highest branches detected and tree length detected for each case ranges from 64.6% to 94.3% and 62.6% to 90.4%, respectively, with an average branches detected and tree length detected of less than 77% and 74%, respectively. Fusing results from the participating algorithms improves the overall result substantially, reaching an average number of branches detected of 84.3% and an average tree length detected of 78.8%, with an average false positive rate of only 1.22%, when all fifteen algorithms[8] were used. Experiments on the inclusion of the results from different algorithms using the sequential forward selection (SFS) procedure show that the tree length of the fused results converges quite rapidly. This indicates that reasonably good results can be obtained by fusing only a subset of the algorithms. Table IV(B) in [8] shows that with a smaller number of algorithms (e.g. using up to 9 algorithms) in the fusion procedure, one can obtain a lower false positive rate at almost the same sensitivity."

Comment 2.9 The branching method described first by Tawhai [4], was first modified by the same group, (Tawhai et al. Journal of Applied Physiology 97: 2310-2321, 2004). Variations of this algorithm have been implemented by Bordas et al. who provided an open-source implementation. But variations have also been presented by Abadi et al (IEEE transactions on medical imaging 37: 693-702, 2018) and Mullally et al. (Ann Biomed Eng 37: 286-300, 2009), and Montesantos et al. PloS one 11: e0168026, 2016). In general, modifications have been made to suit a groups own modelling applications { to claim these changes as an improvement, would require some improved comparison to morphological studies. For the most part, these models perform fairly similarly and are good representations of the airway structure of the lung.

Response: We would like to thank the reviewer for highlighting significant research in the same field. With respect to the definition of the splitting plane, Tawhai et used the direction of the lobar branch and the center of mass of the sampled points subdivision yielding a certain randomness. The same approach was used by mullaly et al. Bordas et al defined the splitting plane by the nodes of the parent branch and the center of mass of the sampled points subdivision. Abadi et al defined any plane containing the center of mass. Montesantos et al used both siblings of of the parent branch to define a splitting point. We also concur that all the aforementioned approaches perform fairly similarly and are good representations of the airway structure of the lung. The motivation for employing PCA is at the space utilization aspect. The following figures give a visual description of the effect. Picking a plane so that the resulting split volumes are less "rectangular", meaning that the sides of the bounding box demonstrate the lowest possible variation, inhibits the appearance of very long branches.

Actual changes implemented: The response for comment 2.9 is provided in comment 2.10.

Comment 2.10

The difference between this and previous methods is a PCA based splitting of seed points, very limited information is given about why one would do this (it seems more complicated than other methods), or what improvements it brings. The algorithm should at the very least be compared to real lung morphometric data, as has been typical in previous studies using similar algorithms. 

Response: The motivation behind employing Principal Component Analysis is that the direction of the eigenvector with the greatest norm indicates the dimension of the data with the greatest variance denoting the direction where more space is available for the branches to grow. Picking a plane so that the resulting split volumes are less "rectangular", meaning that the sides of the bounding box demonstrate the lowest possible variation, inhibits the appearance of very long branches. Although this algorithm might seem complicated, in fact PCA, as a linear transformation, is easy and fast to compute. It has been very popular in many areas solving data normalization and dimensionality reduction problems. In respect to the methods ability to properly reproduce morphometric data, the distribution of branch length and angles as a function of generation, to the best of our knowledge is in agreement with distributions generated by Montesantos et al. Specifically, in the following figure in Montesantos 2016 study presents the airway length as a function of generation and the branching angle as a function of airway generation.

Comment 2.11 It is not clear how the volume of the lung is defined from imaging, or how generated branches are restricted to lie in the lung volume (if at all).

Response: We would like to thank the reviewer for providing us with the opportunity to strengthen weak descriptions of our pipeline. Since the lungs have a lower density than neighbouring tissue and bones, a global threshold is applied between -850 HU and -500 HU to define seed points and perform region growing. For this process, the FAST heterogeneous medical image computing and visualization framework [19] is employed. The result of lung segmentation process is a binary mask visualized in Figure 1. As a next step, we perform further processing of the segmentation result to distinguish left and right lungs. The process is described below: 1. A second region growing takes place starting from a single random point inside any of the segmented region only if all its immediate neighbours bare the same label. 92. To advance the region growing front, all points neighbouring a candidate voxel must not include background voxel. This region is given a new label 3. Steps 1 and 2 are repeated for the other lung volume. 4. The result is an image with three labels background and lung volumes. 5. To distinguish left or right we employ the directed graph extracted from the main airways. As a generic rule, the topological distance between the bifurcations of the first and the second generation is longer in the left lung. Furthermore, for the extended bronchial tree to remain inside the lung volume branches located outside the lung volume are pruned along with any children.

Actual changes implemented: We have modified the segmentation related section 2.1 to include the presented methods. Furthermore, in subsection 2.2 we clarify that any branch located outside the lung volume is pruned along with any children.

Comment 2.12

Given there is presumably a complete 3D representation of the upper airway tree from CT, some

information needs to be given regarding why would want to represent these branches as idealised

tubular structures for CFD?

Response: We would like to thank the reviewer for providing us with the opportunity to highlight the issue of airway morphology. Our approach allows to include a user defined part of the original segmentation to re reconstructed outcome. The extracted 1-dimensional representation is processed to include all the bifurcations located at the end of a given generation so as to facilitate the volume filling algorithm (Please refer to response 2.16 also). Figure 1 shows the result of pruning where all generations after the nth have been pruned. This processed 1-dimensional representation is subsequently used for the bronchial tree extension. The tubular structures were chosen as a model to create a 3D representation of the extended branches in generations for which only the bronchial tree centerlines were available and not a complete 3D representation. Appropriate binding is performed in the interface between upper and lower generations to allow a smooth transition. The first generations segmented from CT retain their original 3D structure unless the user opts for a more simplified representation for the airway tree that could be used to generalize CFD simulations on an average airway tree geometry. The provided code supports both options with the personalized complete 3D representation of the upper airway being the default option. Sampling a point cloud and performing Poisson surface reconstruction produces a watertight surface with smooth transitions and allows us to avoid the definition of special rules in bifurcation surface Furthermore, since the directed graph is extracted where each point on the centerline corresponds to a point on the lung surface it is possible to deform the surface with a custom function or pattern.

Comment 2.13 Figures associated with dataset collection (and some beyond) have no numbers.

Response: We would like to thank the reviewer for indicating us such errors.

Actual changes implemented: The Figure numbers were placed correctly.

Comment 2.14 It is not at all clear what is meant by personalised boundary conditions, or how they are calculated { what is being simulated for which personalised boundary conditions are needed? Or do these boundary conditions relate to tree generation?

Response: Indeed, the personalised boundary conditions refer to the 3D triangular mesh of the bronchial tree extracted for each subject based on the CT images. The first three to four generations are extracted by the segmentation process while a user defined number of subsequent generations is predicted using the branching algorithm. As input, the branching algorithm receives a sampled volume from the lung volume segmentation mask which is personalized for each patient. Furthermore the CT-based extraction of the first generations also reflected the personalized geometry referring to the airway lengths and branching angles of the first generations of a specific subject.

Comment 2.15 To what level is the extended tree generated? The figures suggest that this is not to the level of the acinus as in most previous studies using similar algorithms, but that this is tunable. More information on this would be beneficial, especially in relation to how well the lung is represented it different choices are made.

Response: The number of generation is truly a tunable parameter, thus the tree is extended to any desired generation as long as the produced branches lie inside the lung volume. If n is the desired generation, we set the stopping criteria to 2(n+1) bifurcations. The choice of n depends on the application and the subsequent CFD simulation. Bordas et al. noted that "due to the limitations of CT and the computational difficulty of these (CFD) simulations, studies are typically limited to the first 6-10 generations".Unfortunately at the moment we do not have to groundtruth data that would allow us to quantitatively assess on a subject specific basis the error introduced by new simulated generations for every additional simulated generation. However, the overall distribution of simulated bronchi length and angles, as illustrated in Figure 10 are in agreement with overall statistics observed in the literature for each generation[12].

Comment 2.16 Fig 2 shows one very ling thin branch distending from the main upper airway tree { this does not look anatomical, I would suggest that the algorithm has missed some branches from this main airway, as a very long and thin branch like that is unusual.

Response: This very long branch distending from the main upper airways is most probably a segmentation error. Specifically, the long branch is part of generations 5 to 6 and their children are missing from the segmentation. To deal with such inconsistent segmentation outcomes a postprocessing step is included in the pipeline to remove branches demonstrating uncertainty regarding their generation. Furthermore the processed 1-dimensional representation has to include all the bifurcations located at the end of a given generation so as to facilitate the volume filling algorithm. Figure 1 shows the result of pruning where all generations after the nth have been pruned. This processed 1-dimensional representation is subsequently used for the bronchial tree extension.

Actual changes implemented: We updated Figure 1 so as to include pruning operation.

References

[1] M. H. Tawhai, M. P. Nash, and E. A. Hoffman, \\An imaging-based computational approach to model ventilation distribution and soft-tissue deformation in the ovine lung1," Academic radiology, vol. 13, no. 1, pp. 113{120, 2006. 

[2] C.-L. Lin, M. H. Tawhai, and E. A. Hoffman, \\Multiscale image-based modeling and simulation of gas flow and particle transport in the human lungs," Wiley Interdisciplinary Reviews: Systems Biology and Medicine, vol. 5, no. 5, pp. 643{655, 2013.

[3] M. H. Tawhai, P. Hunter, J. Tschirren, J. Reinhardt, G. McLennan, and E. A. Hoffman, \\Ct-based geometry analysis and finite element models of the human and ovine bronchial tree," Journal of applied physiology, vol. 97, no. 6, pp. 2310{2321, 2004.

[4] R. Bordas, C. Lefevre, B. Veeckmans, J. Pitt-Francis, C. Fetita, C. E. Brightling, D. Kay, S. Siddiqui, and K. S. Burrowes, \\Development and analysis of patient-based complete conducting airways models," PloS one, vol. 10, no. 12, p. e0144105, 2015.

[5] G. R. Mirams, C. J. Arthurs, M. O. Bernabeu, R. Bordas, J. Cooper, A. Corrias, Y. Davit, S.-J. Dunn, A. G. Fletcher, D. G. Harvey et al., \\Chaste: an open source c++ library for computational physiology and biology," PLoS computational biology, vol. 9, no. 3, p. e1002970, 2013.

[6] C. Bauer, H. Bischof, and R. Beichel, \\Segmentation of airways based on gradient vector flow," in International workshop on pulmonary image analysis, Medical image computing and computer assisted intervention. Citeseer, 2009, pp. 191{201.

[7] C. Bauer, T. Pock, H. Bischof, and R. Beichel, \\Airway tree reconstruction based on tube detection," in Proc. of Second International Workshop on Pulmonary Image Analysis, 2009, pp. 203{213.

[8] P. Lo, B. Van Ginneken, J. M. Reinhardt, T. Yavarna, P. A. De Jong, B. Irving, C. Fetita, M. Ortner, R. Pinho, J. Sijbers et al., \\Extraction of airways from ct (exact’09)," IEEE Transactions on Medical Imaging, vol. 31, no. 11, pp. 2093{2107, 2012.

[9] R. Adams and L. Bischof, \\Seeded region growing," IEEE Transactions on pattern analysis and machine intelligence, vol. 16, no. 6, pp. 641{647, 1994.

[10] E. Smistad, A. C. Elster, and F. Lindseth. (2014) Tube segmentation framework. [Online]. Available: https://github.com/smistad/Tube-Segmentation-Framework

[11] E. Smistad, M. Bozorgi, and F. Lindseth. (2015) Fast (framework for heterogeneous medical image computing and visualization). [Online]. Available: https://github.com/smistad/FAST 14

[12] S. Montesantos, I. Katz, M. Pichelin, and G. Caillibotte, \\The creation and statistical evaluation of a deterministic model of the human bronchial tree from hrct images," PLOS one, vol. 11, no. 12, p. e0168026, 2016.

[13] T. Soong, P. Nicolaides, C. Yu, and S. Soong, \\A statistical description of the human tracheobronchial tree geometry," Respiration physiology, vol. 37, no. 2, pp. 161{172, 1979.

[14] I. C. on Radiological Protection, \\Publication 66. human respiratory tract model for radiological protection," Ann. ICRP 24., 1994.

[15] R. Phalen, H. Yeh, G. Schum, and O. Raabe, \\Application of an idealized model to morphometry of the mammalian tracheobronchial tree," The Anatomical Record, vol. 190, no. 2, pp. 167{176, 1978.

[16] H.-C. Yeh and G. Schum, \\Models of human lung airways and their application to inhaled particle deposition," Bulletin of mathematical biology, vol. 42, no. 3, pp. 461{480, 1980.

[17] R. F. Phalen, M. J. Oldham, C. B. Beaucage, T. T. Crocker, and J. Mortensen, \\Postnatal enlargement of human tracheobronchial airways and implications for particle deposition," The Anatomical Record, vol. 212, no. 4, pp. 368{380, 1985.

[18] V. Sauret, P. Halson, I. Brown, J. Fleming, and A. Bailey, \\Study of the three-dimensional geometry of the central conducting airways in man using computed tomographic (ct) images,"Journal of anatomy, vol. 200, no. 2, pp. 123{134, 2002.

[19] E. Smistad, M. Bozorgi, and F. Lindseth, \\Fast: framework for heterogeneous medical image computing and visualization," International Journal of computer assisted radiology and surgery, vol. 10, no. 11, pp. 1811{1822, 2015.

---

## [Decision Letter · Decision Letter 1]

30 Dec 2019

PONE-D-19-24923R1

AVATREE: An open-source computational modelling framework modelling Anatomically Valid Airway TREE conformations

PLOS ONE

Dear Mr Nousias,

Thank you for submitting your manuscript to PLOS ONE. After careful consideration, we feel that it has merit but does not fully meet PLOS ONE’s publication criteria as it currently stands. Therefore, we invite you to submit a revised version of the manuscript that addresses the points raised during the review process.

We would appreciate receiving your revised manuscript by Feb 13 2020 11:59PM. To enhance the reproducibility of your results, we recommend that if applicable you deposit your laboratory protocols in protocols.io, where a protocol can be assigned its own identifier (DOI) such that it can be cited independently in the future. For instructions see: http://journals.plos.org/plosone/s/submission-guidelines#loc-laboratory-protocols

We look forward to receiving your revised manuscript.

Kind regards,

Fang-Bao Tian

Academic Editor

PLOS ONE

Reviewers' comments:

Reviewer's Responses to Questions

**Comments to the Author**

1. If the authors have adequately addressed your comments raised in a previous round of review and you feel that this manuscript is now acceptable for publication, you may indicate that here to bypass the “Comments to the Author” section, enter your conflict of interest statement in the “Confidential to Editor” section, and submit your "Accept" recommendation.

Reviewer #2: (No Response)

2. Is the manuscript technically sound, and do the data support the conclusions?

Reviewer #2: Partly

3. Has the statistical analysis been performed appropriately and rigorously? 

Reviewer #2: N/A

4. Have the authors made all data underlying the findings in their manuscript fully available?

Reviewer #2: Yes

5. Is the manuscript presented in an intelligible fashion and written in standard English?

Reviewer #2: Yes

6. Review Comments to the Author

Reviewer #2: For the most part the authors have satisfied my concerns. However, the need for a comparison between generated tree structures and measured lung morphology remains. The authors have noted that their tree structures are consistent with those generated by Montesantos et al., but don’t provide direct evidence of this in the figures provided. It is left to the reader to go back to that paper and determine how similar the generated trees are. In addition, the Montesantos study is a simulated tree, and it would be most appropriate to compare directly to measurements of airway anatomy. Figures 4-9 of the cited Montesantos et al. paper, shows a comparison between their generated tree, the trees generated by other simulation studies, and, critically, anatomical studies (Horsfield and Cumming, for example). One or two figures like this should be included to verify the model. It would be reasonable for these to be Supplementary Material rather than in the main manuscript, but these data are needed to show how well the model for tree generation performs.

7. PLOS authors have the option to publish the peer review history of their article (what does this mean?). If published, this will include your full peer review and any attached files.

Reviewer #2: No

---

## [Author Response · Author response to Decision Letter 1]

13 Feb 2020

We would like to thank the reviewer for providing us with the opportunity to improve the justification of our approach and the validity of our results. While we were not able to find CT images along with image segmentation from previous studies, thereby not being able to perform a direct volumetric comparison of the extracted trees, we could retrieve analytic results of morphometric analyses of real (not simulated) trees which allowed to validate of our predictions more thoroughly. The following section replaced subsection 3.2 titled "structural modelling" and was included in the main document instead of supplementary material to strengthen the presentation of our results. 

Our simulation framework processes the initial tree centerline and generates a structural estimation given the first three to four available generation and their morphometric characteristics i.e., lengths and diameters. 

To facilitate the comparison with morphometric data, we employed a publicly available dataset provided by Montesantos et al.\\cite{montesantos2016creation} labelled as pone.0168026.s001. For the sake of self-completeness, the authors of \\cite{montesantos2016creation} provided morphometric data extracted from HRCT images acquired at the University Hospital Southampton NHS Foundation Trust as a part of study described in \\cite{fleming2015controlled,majoral2014controlled}. Data from seven healthy subjects and six patients with moderate or persistent asthma were included in the dataset. Asthmatic patients patients were diagnosed exacerbation-free for at least one month and were male non-smokers. 

A Sensation 64 slice HRCT scanner (Siemens, Enlargen, Germany) was utilized to capture 3D images from mouth to the base of the lungs. Subjects were posed in supine position and were instructed to perform slow exhalation. Groundtruth data for the development of bronchial tree models in \\cite{montesantos2016creation} were extracted by Pulmonary Workstation 2 Software including 3 to 4 generations in the upper lobes and 6 to 7 generations in the lower lobes. For each subject, a morphology file includes the total lung volume of the subject lung (in cm^3) and the percent volume per lobe while a translation file contains the airway connectivity, starting from the trachea to the terminal nodes. We used the generated trees from \\cite{montesantos2016creation} to validate our approach and compare our results with relevant literature findings. Specifically, we compared the distributions of diameters, airway lengths and branching angles for each generation and the total number of airways for Horsfield and Strahler orders.

In total 31204 acini were calculated being in agreement with the results reported by \\cite{tawhai2004ct,montesantos2016creation}. Figures 10 and 11 present a comparison in terms of the number of airways for each level of Strahler and Horsfield orders. This comparison confirms that our model comes into agreement with pone.0168026.s001. Furthermore, distributions of airway lengths, branching angles and diameters were plotted for each generation, for AVATREE and pone.0168026.s001\\cite{montesantos2016creation}.

Airway lengths maintain the same exponential decay pattern for both models. Differences appear in generations 1 to 4 that are distinctively defined by body size and anatomical features. The distribution of branching angles of our model is also confirmed by pone.0168026.s001\\cite{montesantos2016creation} maintaining a nearly linear decay with a very small rate. The distributions of diameters per generation are also observed to follow an exponential decay pattern. Both our model and pone.0168026.s001\\cite{montesantos2016creation} decay similarly after generation 4 validating the morphometric characteristics of the airway trees generated by our approach. Figures 12 to 14 present the distribution of airway length, branching angle and diameter for each generation for AVATREE and for pone.0168026.s001 \\cite{montesantos2016creation}.

Table 1 presents and overview of quantitative macroscopic figures for AVATREE and relevant studies.

Branching ratios (RB_H,RB_S), diameter ratios (RD_H,RD_S) and length ratios RL_H,RL_S) were calculated for Strahler and Horsfield ratios denoted as *_S and *_H respectively. Specifically, RB_H,RD_H and RL_H were calculated equal to RB_H=1.74, RD_H=1.259 and RL_H=1.26+- 1.01. Montesantos et al.\\cite{montesantos2016creation} reported RB_H=1.56, RD_H=1.115 and RL_H=1.13 respectively. Additionally, RB_S,RD_S and RL_S were calculated equal to RB_S=2.35, RD_S=1.25 and RL_S=1.23+- 1.02 and are close to the figures provides by relative studies \\cite{horsfield1986morphometry,montesantos2016creation} as Table 1 reveals. Likewise, rate of decline for diameters per generation RD was calculated to RD=0.83 +- 0.21, being in agreement to \\cite{montesantos2016creation}. 

Finally, average branching angle theta for our model was calculated to 32.44+-28.95 comparable to \\cite{montesantos2016creation} reporting a theta equal to 42.1+- 21.4.

---

## [Decision Letter · Decision Letter 2]

26 Feb 2020

AVATREE: An open-source computational modelling framework modelling Anatomically Valid Airway TREE conformations

PONE-D-19-24923R2

Dear Dr. Nousias,

We are pleased to inform you that your manuscript has been judged scientifically suitable for publication and will be formally accepted for publication once it complies with all outstanding technical requirements.

With kind regards,

Fang-Bao Tian

Academic Editor

PLOS ONE

Additional Editor Comments (optional):

Reviewers' comments:

Reviewer's Responses to Questions

**Comments to the Author**

1. If the authors have adequately addressed your comments raised in a previous round of review and you feel that this manuscript is now acceptable for publication, you may indicate that here to bypass the “Comments to the Author” section, enter your conflict of interest statement in the “Confidential to Editor” section, and submit your "Accept" recommendation.

Reviewer #2: All comments have been addressed

2. Is the manuscript technically sound, and do the data support the conclusions?

Reviewer #2: Yes

3. Has the statistical analysis been performed appropriately and rigorously? 

Reviewer #2: Yes

4. Have the authors made all data underlying the findings in their manuscript fully available?

Reviewer #2: Yes

5. Is the manuscript presented in an intelligible fashion and written in standard English?

Reviewer #2: Yes

6. Review Comments to the Author

Reviewer #2: The authors have satisfied my concerns raised in the last round of review. Thank you for adding the statistical information on the trees, which will help readers interpret the paper.

7. PLOS authors have the option to publish the peer review history of their article (what does this mean?). If published, this will include your full peer review and any attached files.

Reviewer #2: No

---

## [Editor Report · Acceptance letter]

18 Mar 2020

PONE-D-19-24923R2 

AVATREE: An open-source computational modelling framework modelling Anatomically Valid Airway TREE conformations 

Dear Dr. Nousias:

I am pleased to inform you that your manuscript has been deemed suitable for publication in PLOS ONE. Congratulations! Your manuscript is now with our production department. 

With kind regards,

on behalf of

Dr. Fang-Bao Tian 

Academic Editor

PLOS ONE